# Perivascular cell-specific knockout of the stem cell pluripotency gene Oct4 inhibits angiogenesis

Daniel L. Hess[1,2], Molly R. Kelly-Goss[1,3], Olga A. Cherepanova[4], Anh T. Nguyen[1], Richard A. Baylis [1,2], Svyatoslav Tkachenko[5], Brian H. Annex[1,6], Shayn M. Peirce[1,3] & Gary K. Owens[1,7]

The stem cell pluripotency factor Oct4 serves a critical protective role during atherosclerotic plaque development by promoting smooth muscle cell (SMC) investment. Here, we show using Myh11-CreER[T2] lineage-tracing with inducible SMC and pericyte (SMC-P) knockout of *Oct4* that Oct4 regulates perivascular cell migration and recruitment during angiogenesis. Knockout of *Oct4* in perivascular cells significantly impairs perivascular cell migration, increases perivascular cell death, delays endothelial cell migration, and promotes vascular leakage following corneal angiogenic stimulus. Knockout of *Oct4* in perivascular cells also impairs perfusion recovery and decreases angiogenesis following hindlimb ischemia. Transcriptomic analyses demonstrate that expression of the migratory gene Slit3 is reduced following loss of Oct4 in cultured SMCs, and in Oct4-deficient perivascular cells in ischemic hindlimb muscle. Together, these results provide evidence that Oct4 plays an essential role within perivascular cells in injury- and hypoxia-induced angiogenesis.

[1] Robert M. Berne Cardiovascular Research Center, University of Virginia-School of Medicine, 415 Lane Road, Suite 1010, Charlottesville, VA 22908, USA. [2] Department of Biochemistry and Molecular Genetics, University of Virginia-School of Medicine, Charlottesville, VA 22908, USA. [3] Department of Biomedical Engineering, University of Virginia-School of Medicine, Charlottesville, VA 22908, USA. [4] Lerner Research Institute, 9500 Euclid Avenue, NB50, Cleveland, OH 44195, USA. [5] Lerner Research Institute, 9500 Euclid Avenue, JJN3-01, Cleveland, OH 44195, USA. [6] Department of Medicine, Cardiovascular Medicine, University of Virginia, Charlottesville, VA 22908, USA. [7] Department of Molecular Physiology and Biological Physics, University of Virginia-School of Medicine, Charlottesville, VA 22908, USA. These authors contributed equally: Daniel L. Hess, Molly R. Kelly-Goss. Correspondence and requests for materials should be addressed to G.K.O. (email: gko@virginia.edu)

Octamer-binding transcription factor 4 (Oct4) is a stem cell pluripotency gene critical for maintenance of pluripotency in the inner cell mass of the blastocyst[1]. Oct4 expression is tightly regulated during embryogenesis and declines during germ layer specification through epigenetic repression via DNA and histone methylation[2]. The long-standing dogma in the field was that this epigenetic silencing is permanent in all adult somatic cells[2–4]. Contrary to dogma, a number of studies have reported Oct4 expression in a variety of stem and progenitor cell populations[3]. However, these studies failed to provide evidence that Oct4 had a functional role in these cells, and were viewed with extensive skepticism due to a number of potential false positives associated with Oct4 transcript and protein detection, including the presence of multiple Oct4 non-pluripotent isoforms and pseudogenes[3]. Our lab also detected Oct4 expression in somatic cells, namely in smooth muscle cells (SMC) in mouse and human atherosclerotic lesions, and utilized a murine genetic loss-of-function approach to conditionally and specifically delete the pluripotency isoform of Oct4 in SMC[5]. We found that Oct4 plays a critical protective role in SMC, in that Oct4 deletion impaired investment of SMC into both the lesion and fibrous cap during atherosclerosis, and was associated with increased atherosclerotic burden and decreased indices of plaque stability[5]. Of major significance, this was the first direct evidence that Oct4 plays a functional role in any somatic cell. Therefore, despite epigenetic silencing during gastrulation, the Oct4 locus evolved the capacity to be reactivated and serve a function in SMC.

Interestingly, the clinical manifestations of atherosclerosis, including thromboembolic complications, such as stroke and myocardial infarction, affect individuals well after their reproductive years, and as such there would have been no selective pressure for Oct4 to evolve a role to combat atherosclerosis development or end stage complications. Therefore, Oct4 re-activation in SMC may be an anomaly unique to pathological states as has been surmised by numerous investigators claiming it is re-activated in cancer stem cells[6]. Alternatively, Oct4 may have evolved a protective role in SMC to enhance processes critical for survival and reproductive success and only secondarily developed a role during atherosclerosis development. Angiogenesis, or the growth of new blood vessels from a pre-existing vasculature, is essential for survival and reproduction, as it is responsible for supply of oxygen and nutrients[7,8]. Since angiogenesis requires perivascular cell investment for the formation of functional vascular networks, we postulated that Oct4 evolved to play a critical role in this process.

Angiogenesis requires coordinated migration of the two major cell types of the blood vessel wall: (1) endothelial cells (EC), which line the inner lumen and (2) perivascular cells (SMC and pericytes), which envelop EC. In general, SMC concentrically wrap arteries, arterioles, veins, and venules which have diameters >10 μm, while pericytes extend longitudinally along capillaries <10 μm in diameter. Despite these distinct anatomical differences, SMC and pericytes often express many common proteins including ACTA2, MYH11, and PDGFR-β, which vary in expression across different vascular beds under both normal and pathologic conditions[9]. Indeed, no marker or set of markers has been able to unequivocally distinguish SMC from pericytes[9]. For this reason, and due to their shared contributions to angiogenic perivascular populations[10], we henceforth refer to them together as SMC and pericytes (SMC-P).

During angiogenesis, EC and SMC-P communication is essential for new blood vessel formation[11]. Perivascular cell-selective knockout of Ephrin-B2[12], CD146[13], Tie2[14], and VEGFR1[15] result in impaired angiogenesis, including defective SMC-P investment of EC tubes and increased vascular leak. However, despite these insights into mechanisms by which perivascular cells contribute to functional vascular networks, dysregulated angiogenesis continues to play a major role in numerous disease processes including cardiovascular disease and cancer, the two leading causes of death in the US[16]. Numerous cardiovascular clinical trials for peripheral arterial disease (PAD), heart failure, and stroke have sought to augment angiogenesis by administering growth factors that promote EC proliferation and migration (e.g. VEGF and bFGF) as a means to increase tissue perfusion and recovery[17–20]. However, the majority of these studies have failed to meet their primary endpoints, which is at least partially due to EC growth without coordinated SMC-P migration and investment[21]. Thus, further insight into the mechanisms regulating perivascular cell involvement in angiogenesis is critical for developing therapies to modulate functional vascular growth and remodeling.

Herein, we test the hypothesis that SMC-P derived Oct4 is essential for angiogenesis following in vivo injury, including corneal burn and hindlimb ischemia (HLI). We demonstrate that SMC-P conditional deletion of the stem cell pluripotency factor Oct4 results in markedly impaired angiogenesis, at least in part through defective SMC-P migration leading to increased vascular leak. This study identifies a pathway implicating the key stem cell factor Oct4 in regulation of perivascular cell investment and vascular network remodeling following tissue injury or hypoxia.

## Results

**Myh11-CreER^T2 eYFP labeled SMC and a subset of pericytes**. We previously developed a Myh11-CreER[T2] ROSA floxed STOP eYFP inducible lineage tracing mouse that specifically labels >95% of SMC within large conduit arteries with eYFP following tamoxifen injection between 6 and 8 weeks of age[22]. Recently, using this mouse, we demonstrated that a subset of eYFP+ cells in the pre-metastatic lungs express the pericyte markers NG2 and PDGFR-β[23]. To determine whether the Myh11-CreER[T2] system also labels pericytes in a number of other microvascular beds, we crossed our Myh11-CreER[T2] ROSA floxed STOP eYFP mice to NG2-DsRED reporter mice to generate Myh11-CreER[T2] ROSA eYFP NG2-DsRED mice (Fig. 1a). We examined whole mounts of the retina, which has the highest pericyte density in the body[11], and observed eYFP+ cells surrounding isolectin + EC tubes, including those of capillary size diameter (Fig. 1b). There was co-localization of eYFP and NG2-DsRed in pericytes surrounding isolectin + EC tubes, demonstrating eYFP efficiently labels NG2+ pericytes in the retina (Fig. 1c). We also examined the limbal vasculature of the cornea and observed labeling of NG2-DsRED+ cells with eYFP (Fig. 1d). We next used our Myh11-CreER[T2] ROSA eYFP lineage tracing mice (Fig. 1e) to quantify the efficiency of eYFP labeling of pericytes in calf muscle vasculature. We found that ~80% of eYFP+ cells surrounding CD31+ capillaries express multiple pericyte markers, including NG2 and PDGFR-β (Fig. 1f–h). Over 90% of NG2+ cells, and 80% of PDGFR-β+ cells, express eYFP (Fig. 1i). Taken together, our results demonstrate that the Myh11-CreER[T2] system labels both SMC[22] and a large subset of pericytes within multiple tissues. Using Myh11-CreER[T2], we can therefore conditionally delete Oct4 in both SMC and pericytes to test for a functional role during angiogenesis following injury.

**SMC-P Oct4 knockout had no effect on baseline phenotype**. To knock out Oct4 and determine its role in Myh11-expressing SMC-P, we injected Myh11-CreER[T2] ROSA floxed STOP eYFP Oct4^WT/WT and Myh11-CreER[T2] ROSA floxed STOP eYFP Oct4^FL/FL male littermate mice with tamoxifen from 6 to 8 weeks of age, as previously described[5]. This induces permanent lineage tagging of Myh11-expressing cells with eYFP, without and with

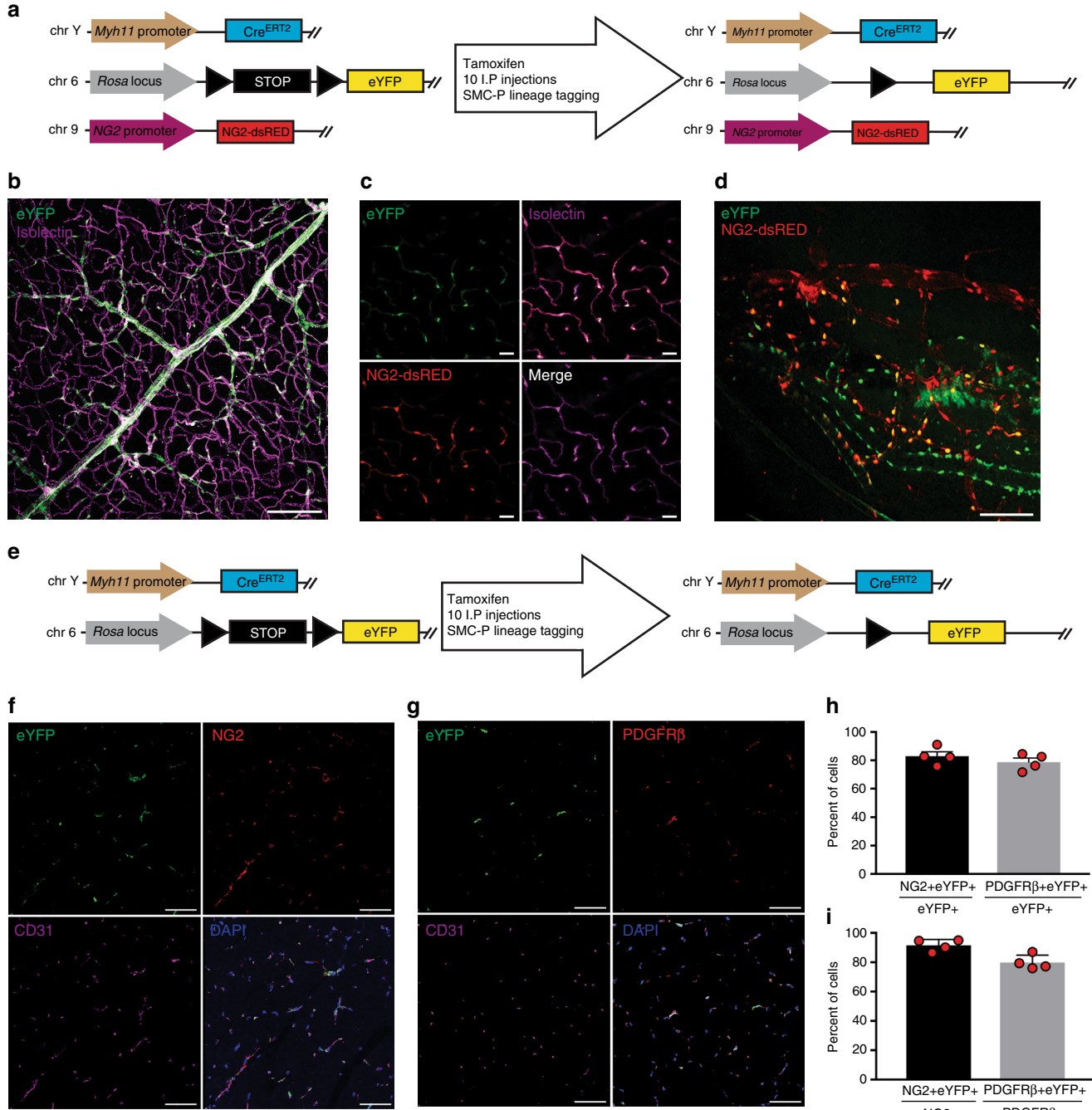

**Fig. 1** Myh11-CreER[T2] ROSA eYFP efficiently labeled SMC and a large subset of pericytes in multiple microvascular tissue beds. **a** Schematic showing crossing of Myh11-CreER[T2] ROSA floxed STOP eYFP mice with NG2-DsRED mice plus tamoxifen injection to generate NG2-DsRED Myh11-CreER[T2] ROSA eYFP mice. **b** and **c** Imaging of retina whole mounts for eYFP, NG2-DsRED, and isolectin. Scale bar in **b** = 100 μm. Scale bars in **c** = 20 μm. **d** Intravital microscopy of cornea limbal vasculature for eYFP and NG2-DsRED. Scale bar = 50 μm. **e** Schematic showing Myh11-CreER[T2] ROSA eYFP mice. **f** and **g** Co-staining of uninjured calf muscle cross sections from Oct4[SMC-P WT/WT] mice for DAPI, eYFP, and NG2 (**f**) or PDGFR-β (**g**). Scale bars = 50 μm. **h** and **i** Quantification of percentages of dual positive cells within calf muscle ($n = 4$ mice). Values = mean ± s.e.m

*Oct4* knockout, respectively, exclusively in SMC-P (Fig. 2a). Henceforth, we refer to them as *Oct4*[SMC-P WT/WT] and *Oct4*[SMC-P Δ/Δ] mice. To ensure *Oct4* knockout, we tested recombination at the Oct4 locus in a number of tissues, including aorta, liver, lung, diaphragm, skeletal muscle, and blood. Following tamoxifen administration, we observed *Oct4* recombination in *Oct4*[SMC-P Δ/Δ] tissues but not in *Oct4*[SMC-P WT/WT] tissues. Additionally, no *Oct4* recombination was observed in blood, which we previously showed contains no eYFP+ cells

(Supplementary Figure 1b)[24]. As further validation, we sorted eYFP+ and eYFP− cells from calf muscle of *Oct4*[SMC-P WT/WT] and *Oct4*[SMC-P Δ/Δ] mice (Supplementary Figure 1c). We observed *Oct4* recombination only in the eYFP+ population sorted from *Oct4*[SMC-P Δ/Δ] mice (Supplementary Figure 1d).

We then determined whether loss of *Oct4* in Myh11-expressing SMC-P results in any effect prior to an angiogenic stimulus. Twelve to eighteen-week-old *Oct4*[SMC-P Δ/Δ] mice did not show any significant changes in total body weight, blood pressure, or

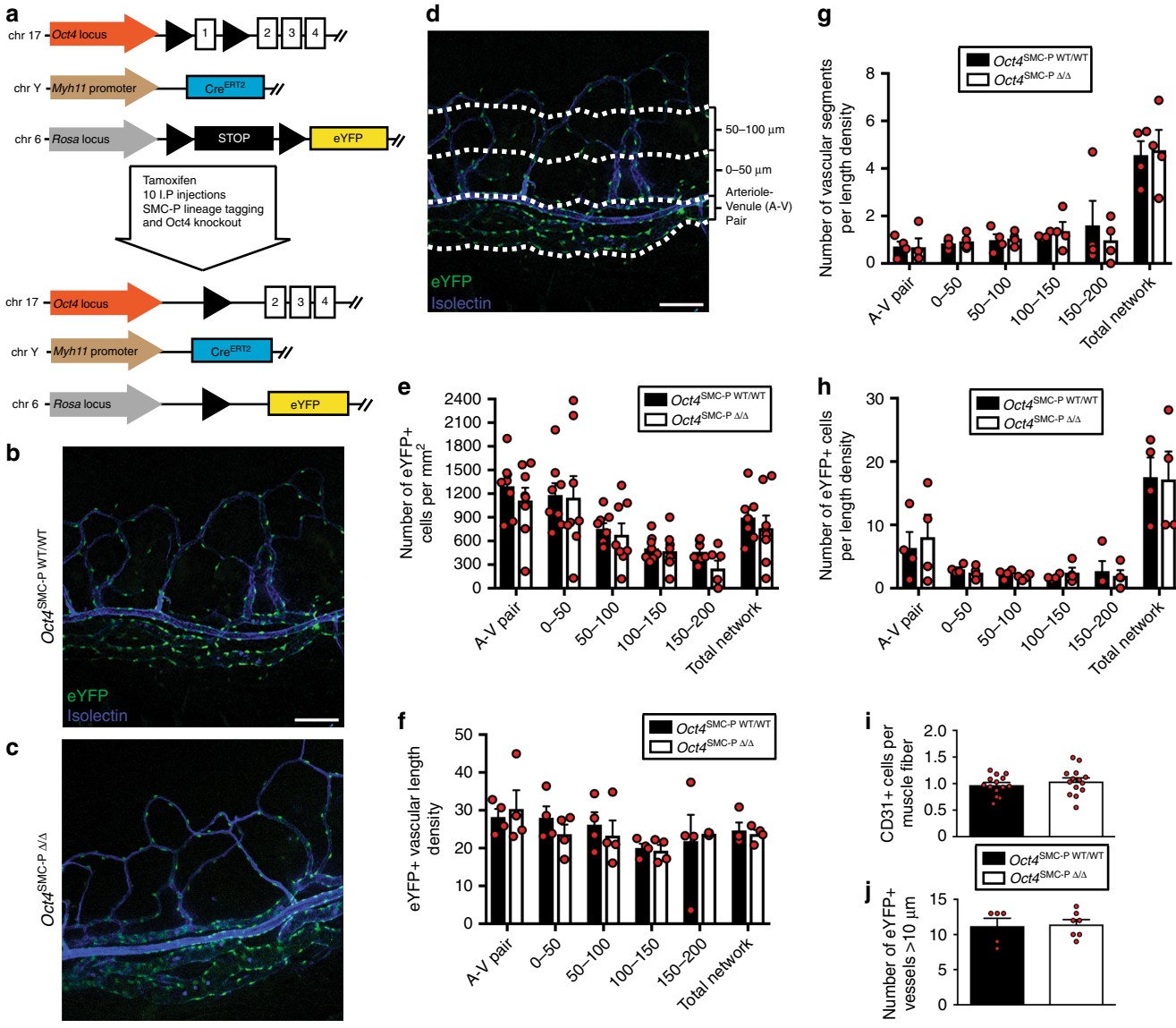

**Fig. 2** There were no differences in the corneal or hindlimb vasculature at baseline following SMC-P knockout of Oct4. **a** Myh11-CreER[T2] ROSA floxed STOP eYFP Oct4[WT/WT] and Myh11-CreER[T2] ROSA floxed STOP eYFP Oct4[FL/FL] male littermate mice were injected with tamoxifen (10 intraperitoneal injections) from 6 to 8 weeks of age to induce simultaneous lineage tagging, without or with Oct4 KO, respectively. For simplicity, we refer to them henceforth as Oct4[SMC-P WT/WT] and Oct4[SMC-P Δ/Δ], respectively. **b**, **c** Representative intravital confocal images of Oct4[SMC-P WT/WT] and Oct4[SMC-P Δ/Δ] corneas showing native eYFP (green) and perfused isolectin (blue). Scale bar = 50 μm. **d** Prior to injury, there is limbal vasculature consisting of a main arteriole–venule (A–V) pair at the base of the vascular network, as well as vessels extending ~200 μm away from the A–V pair toward the center of the cornea. To more rigorously quantify number of cells in the limbal vasculature, we divided this region in to 50 μm regions, where 50 μm was measured from the edge of the A–V pair towards the center of the cornea along the total width of the montaged image. Scale bar = 50 μm. **e–h** Quantification of eYFP+ cell density (**e**), eYFP+ vascular length density (**f**), eYFP+ vascular segments per length density (**g**), and number of eYFP+ cells per length density (**h**) in Oct4[SMC-P WT/WT] and Oct4[SMC-P Δ/Δ] corneas immediately prior to injury [n = 8 WT, 8 KO (**e**); n = 4 WT, 4 KO (**f–h**)]. **i–j** Quantification of the number of CD31+ cells per muscle fiber (**i**) and number of eYFP+ vessels >10 μm diameter (**j**) in hindlimb muscle of Oct4[SMC-P WT/WT] and Oct4[SMC-P Δ/Δ] mice [n = 14 WT, 14 KO (**i**); n = 5 WT, 7 KO (**j**)]. Values = mean ± s.e.m. Statistics were performed using two-way ANOVA (**e–h**), unpaired two-tailed t-test (**i**), or Mann–Whitney U-test (**j**)

heart rate, compared to littermate controls (Supplementary Figure 2). We administered Alexaflour-647-labeled isolectin GS-IB4 (lectin) via retro-orbital injection 10 min prior to imaging in order to visualize perfusion-competent blood vessels. We then used intravital confocal microscopy to image the corneal limbal vascular bed of Oct4[SMC-P WT/WT] and Oct4[SMC-P Δ/Δ] mice for injected isolectin and native eYFP expression (Fig. 2b, c). The corneal limbal vasculature consists of an arteriole–venule (A–V) pair encircling the cornea circumference, as well as a vascular network that extends ~200 μm away from the main A–V pair

towards the center of the cornea. We divided the limbal vasculature into 50 μm regions and assessed a number of vascular parameters at baseline, i.e. immediately prior to injury (Fig. 2d). We observed no significant differences in eYFP+ vascular length density, number of eYFP+ vascular segments, eYFP+ cell coverage, or eYFP+ cell density between Oct4[SMC-P WT/WT] and Oct4[SMC-P Δ/Δ] corneas immediately prior to injury (Fig. 2e–h). We then assessed the baseline vasculature in hindlimb muscle of Oct4[SMC-P WT/WT] and Oct4[SMC-P Δ/Δ] mice. We saw no difference in capillary density or number of eYFP+ vessels within the

hindlimb (Fig. 2i, j). Taken together, results show that, in the absence of injury, conditional knockout of *Oct4* within perivascular cells has no effect on corneal limbal or hindlimb vasculature.

**SMC-P knockout of Oct4 impaired corneal angiogenesis**. To test whether Oct4 plays a functional role in SMC-P during angiogenesis following injury, we subjected twelve to eighteen-week-old *Oct4*$^{SMC-P\ WT/WT}$ and *Oct4*$^{SMC-P\ \Delta/\Delta}$ mice to corneal alkali burn injury (Fig. 3a), as previously described[25]. Following corneal burn, the limbal vasculature gives rise to angiogenic sprouts that extend towards the alkali burn in the center of the cornea (Supplementary Figure 3). We used intravital confocal microscopy to visualize isolectin+ vessel lumens and eYFP+ cells during angiogenesis towards the burn site at days 3, 7, and 21 post-corneal burn. We divided the network area into defined regions, including the A–V pair and 50 μm regions extending away from the A–V pair towards the burn (Supplementary Figure 3). We then calculated the number of eYFP+ cells within each region and normalized to area.

By day 3 post-corneal burn, the distribution of eYFP+ cells throughout the remodeling vascular network was significantly different in *Oct4*$^{SMC-P\ \Delta/\Delta}$ corneas relative to *Oct4*$^{SMC-P\ WT/WT}$ corneas. Moreover, there was a significant decrease in eYFP+ cell density throughout the angiogenic network in *Oct4*$^{SMC-P\ \Delta/\Delta}$ corneas (Supplementary Figure 4a). At day 7 post-corneal burn, eYFP+ cell distribution throughout the remodeling corneal network was also significantly different between *Oct4*$^{SMC-P\ \Delta/\Delta}$ and *Oct4*$^{SMC-P\ WT/WT}$ corneas. Specifically, in *Oct4*$^{SMC-P\ WT/WT}$ corneas at day 7 post-burn, eYFP+ cells migrated away from the limbal vasculature to form a dense neovascular area extending ~600 μm away from the limbal A–V pair. In *Oct4*$^{SMC-P\ \Delta/\Delta}$ corneas, however, there were very few eYFP+ cells that had migrated >350 μm away from the limbal A–V pair (Fig. 3d, e). At day 21 post-corneal burn, eYFP+ cell distribution remained significantly different across genotypes. In *Oct4*$^{SMC-P\ WT/WT}$ corneas, numerous eYFP+ cells migrated out to 900 μm away from the A–V pair. In contrast, in *Oct4*$^{SMC-P\ \Delta/\Delta}$ corneas, there were very few eYFP+ cells that had migrated >450 μm away from the limbus (Fig. 3f, g). The altered distribution of eYFP+ cells throughout the remodeling angiogenic network at all time points post-burn suggests eYFP+ cells lacking Oct4 may be unable to effectively migrate following corneal burn.

In addition to significantly altered distribution of eYFP+ cells following loss of Oct4, there was also significantly decreased eYFP+ cell density throughout the entire network area in *Oct4*$^{SMC-P\ \Delta/\Delta}$ corneas (Fig. 3h). This suggests that, in the absence of Oct4, eYFP+ cells undergo increased cell death and/or decreased proliferation following corneal burn, resulting in decreased cell density. To test whether eYFP+ cells lacking Oct4 are more susceptible to apoptosis following burn, we measured apoptotic DNA fragmentation with TUNEL. At day 1 post-injury, *Oct4*$^{SMC-P\ \Delta/\Delta}$ corneas had significantly increased ratios of TUNEL+ eYFP+ cells to total eYFP+ cells, when compared to *Oct4*$^{SMC-P\ WT/WT}$ corneas. However, at days 2 and 5 post-injury, there was no significant difference in TUNEL staining of eYFP+ cells between *Oct4*$^{SMC-P\ WT/WT}$ and *Oct4*$^{SMC-P\ \Delta/\Delta}$ corneas (Supplementary Figure 4b). To determine whether eYFP+ cell proliferation was altered as a function of loss of Oct4, we stained for Ki67 at days 2 and 5 post-corneal burn and found ~18% and 35% of eYFP+ cells, respectively, were in active phases of the cell cycle. However, rates of eYFP+ cell proliferation were similar between *Oct4*$^{SMC-P\ WT/WT}$ and *Oct4*$^{SMC-P\ \Delta/\Delta}$ corneas (Supplementary Figure 4c). Therefore, *Oct4* knockout leads to increased apoptosis of eYFP+ cells at day 1 post-burn, contributing to significantly reduced eYFP+ cell density at

day 3 post-burn. *Oct4* knockout also reduced the distance of eYFP+ cells from the corneal limbus, likely due in part to impaired eYFP+ cell migration. However, we cannot rule out additional changes in eYFP+ proliferation or apoptosis that we failed to detect. Taken together, these data suggest that SMC-P Oct4 is critical for effective angiogenesis following corneal burn.

**SMC-P knockout of Oct4 increased vascular leakage**. We then tested if changes in eYFP+ cells due to SMC-P knockout of *Oct4* played a role in the patency and functionality of the angiogenic vessels, as well as the overall health of the healing cornea post-alkali burn. Therefore, we examined vascular leakage, incidence of hemorrhage, and cornea tissue damage.

To assess vascular leakage, we administered a retro-orbital injection of 70 kDa rhodamine–dextran into the contralateral eye immediately (< 3 min) prior to live confocal imaging at day 3 post-corneal burn. We observed very little dextran leak in *Oct4*$^{SMC-P\ WT/WT}$ corneas (Fig. 4a; Supplementary Movie 1). However, in *Oct4*$^{SMC-P\ \Delta/\Delta}$ corneas, there was extensive accumulation of 70 kDa rhodamine–dextran in the interstitial space, indicative of vascular leakage (Fig. 4a; Supplementary Movie 2). Indeed, when we quantified fluorescence intensity of dextran in regions outside the vasculature, vascular leak was increased more than four-fold in *Oct4*$^{SMC-P\ \Delta/\Delta}$ corneas compared to *Oct4*$^{SMC-P\ WT/WT}$ corneas (Fig. 4b).

To evaluate the extent of hemorrhaging in *Oct4*$^{SMC-P\ \Delta/\Delta}$ corneas, we took bright field images of *Oct4*$^{SMC-P\ WT/WT}$ and *Oct4*$^{SMC-P\ \Delta/\Delta}$ corneas at days 0, 3, 7, and 21 post-corneal burn and quantified the number of hemorrhages, or areas of blood not confined to the vasculature, using a semi-quantitative scoring system based on a clinical scale (Table 1). The scoring also assessed the health of the corneal tissue, i.e. whether it was healthy, torn, or ruptured, which directly correlated with the extent of hemorrhaging. We observed increased hemorrhaging in *Oct4*$^{SMC-P\ \Delta/\Delta}$ mice, indicative of increased vascular leak and consistent with reduced eYFP+ cell migration and investment of nascent EC tubes (Fig. 4c).

**SMC-P knockout of Oct4 resulted in delayed EC migration**. Given the important role of SMC-P/EC crosstalk during angiogenesis, we asked whether SMC-P *Oct4* knockout results in secondary changes in EC. Thus, we performed whole mount immunostaining of post-burn corneas for the EC marker CD31. In *Oct4*$^{SMC-P\ \Delta/\Delta}$ corneas at day 3 post-burn, there was a significant decrease in CD31+ migration distance, defined as the maximum distance CD31+ cells extended away from the A–V pair (Fig. 5a–c). However, sprout formation at day 3 was unaffected by loss of Oct4 in SMC-P (Fig. 5d). By day 7 post-corneal burn, CD31+ distance from the A–V pair was not significantly different between *Oct4*$^{SMC-P\ WT/WT}$ and *Oct4*$^{SMC-P\ \Delta/\Delta}$ corneas (Fig. 5e). We then assessed EC apoptosis (Fig. 5f) and proliferation (Fig. 5g) at days 2 and 5, which could possibly contribute to the changes at days 3 and 7, respectively, but observed no significant difference between genotypes. Taken together, results indicate that loss of Oct4 within perivascular cells results in a transient delay in EC migration at day 3 post-burn that is compensated for by day 7 post-burn.

This compensation in EC migration, along with no other detectable changes in EC function despite significant eYFP+ cell dropout, raised the possibility that eYFP− populations of SMC-P may compensate for the loss of eYFP+ cells following *Oct4* knockout. To test this hypothesis, we stained eye cross-sections at day 5 post-burn for CD31, eYFP, and NG2, an additional pericyte marker. In *Oct4*$^{SMC-P\ WT/WT}$ corneas, eYFP+ and NG2+ cells invested CD31+ EC tubes throughout the proximal and distal

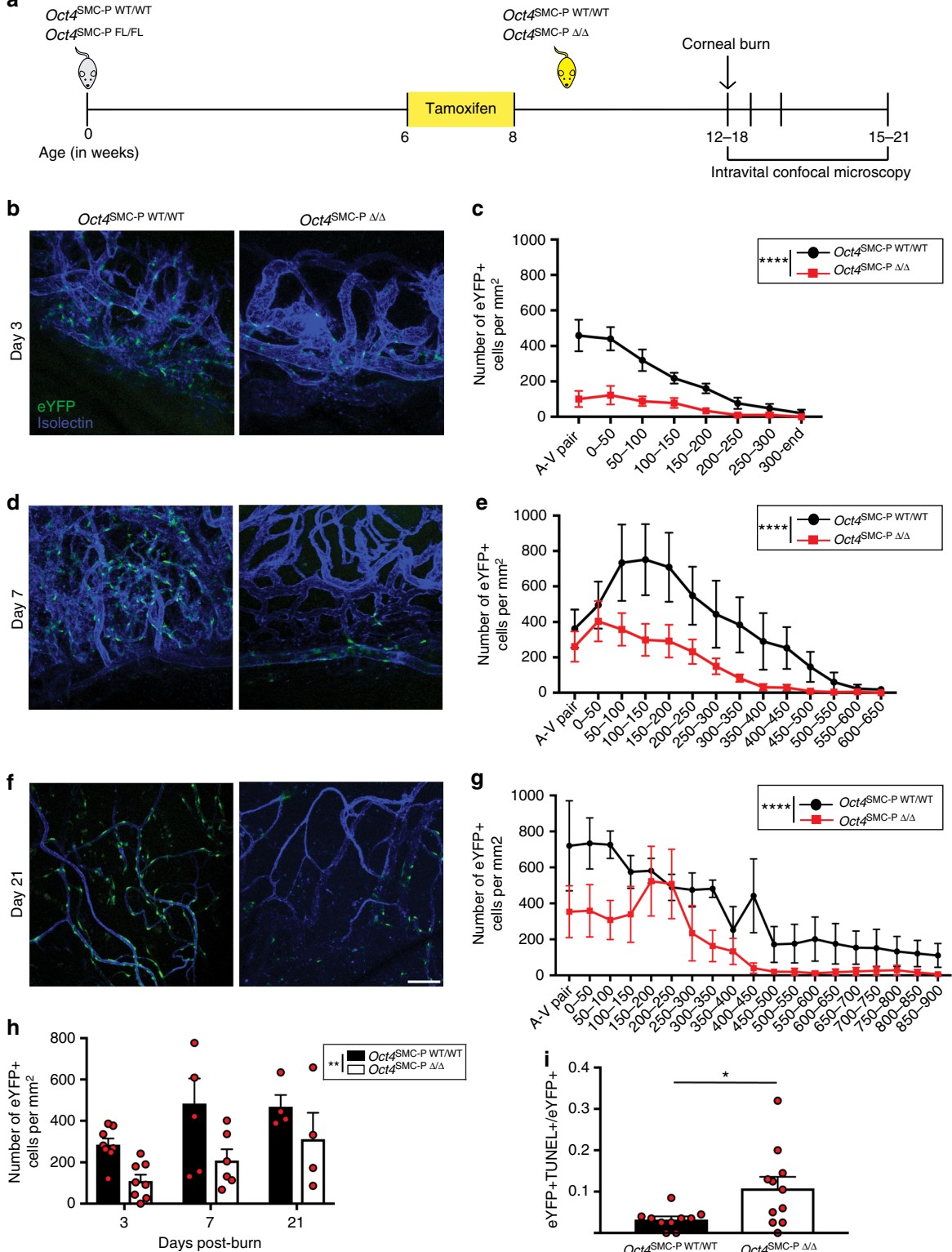

**Fig. 3** SMC-P conditional knockout of Oct4 resulted in impaired migration and increased cell death of eYFP+ cells following corneal burn. **a** Schematic outlining experimental design. **b–g** Representative intravital confocal microscopy images for native eYFP (green) and perfused isolectin (blue) and quantification of the number of eYFP+ cells per area at days 3 (**b**, **c**), 7 (**d**, **e**), and 21 (**f**, **g**) post-corneal burn. Scale bar = 50 μm [$n = 8$ WT, 8 KO (**c**); $n = 6$ WT, 6 KO (**e**); $n = 4$ WT, 4 KO (**g**)]. **h** Quantification of the number of eYFP+ cells throughout the entire network area at days 3, 7, and 21 post-burn. **i** Quantification of the ratio of TUNEL+ eYFP+ cells to total eYFP+ cells in cornea vasculature at day 1 ($n = 10$ WT, 11 KO) post-burn. Values = mean ± s.e.m. Statistics were performed using unpaired two-tailed $t$-test (**i**) or two-way ANOVA (**c**, **e**, **g**, **h**). *$P < 0.05$, **$P < 0.01$, ****$P < 0.0001$

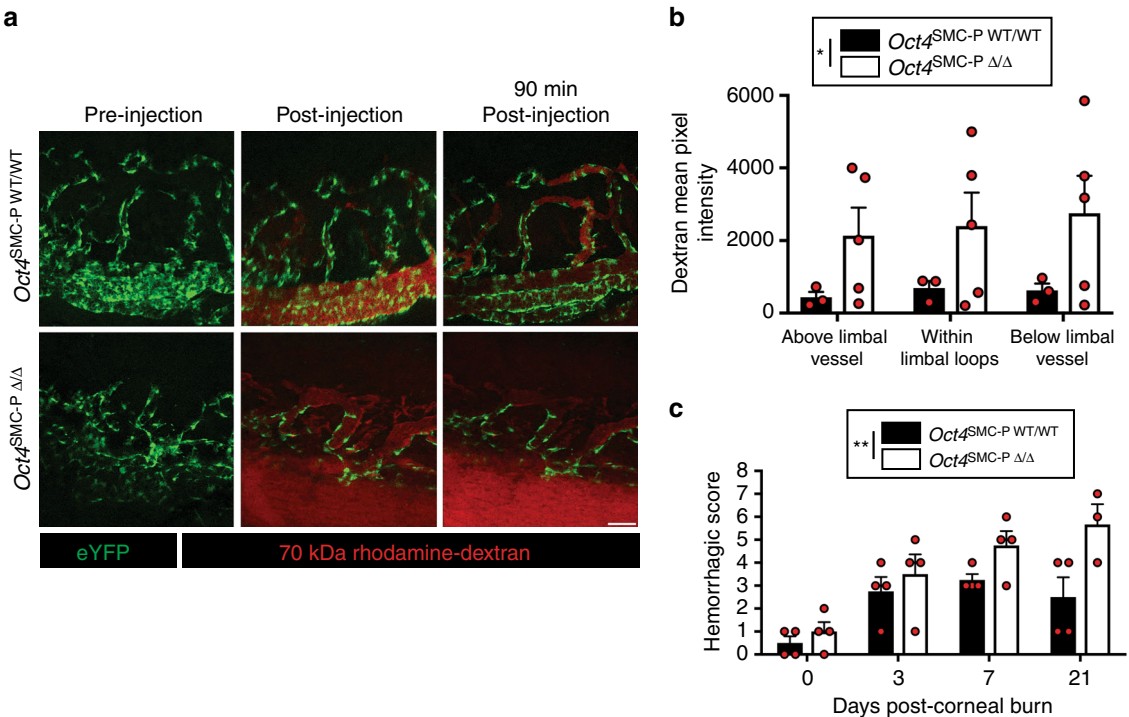

**Fig. 4** SMC-P knockout of Oct4 resulted in increased vascular leak and hemorrhaging following corneal burn. **a** Representative still-frame images of Oct4[SMC-P WT/WT] and Oct4[SMC-P Δ/Δ] corneas pre-injection, immediately post-injection, and 90 min post-injection of 70 kDa rhodamine–dextran (red). Native eYFP expression is shown in green. **b** Quantification of dextran mean pixel intensity of defined regions throughout the vascular network ($n = 3$ WT, 5 KO). Scale bar = 50 μm. **c** Quantification of hemorrhaging and cornea integrity at each time point post-corneal burn ($n = 4$ WT, 4 KO for D0, D3, D7; $n = 4$ WT, 3 KO for D21). Values = mean ± s.e.m. Statistics were performed using two-way ANOVA (**b**, **c**). *$P < 0.05$ and **$P < 0.01$

**Table 1 Scale used to quantify extent of hemorrhaging and health of corneal tissue following burn. Scale is based on the number of hemorrhages surrounding the corneal circumference as well as the integrity of the corneal tissue**

| Hemorrhagic score | Number of hemorrhages | Corneal tissue |
|---|---|---|
| 0 | 0 | Healthy |
| 1 | 1 | Healthy |
| 2 | 2–3 | Healthy |
| 3 | 2–3 | Disrupted, Torn |
| 4 | Continuous | Healthy |
| 5 | 3–4 | Disrupted, Torn |
| 6 | Continuous | Disrupted, Torn |
| 7 | Continuous | Total rupture |

corneal vascular network (Supplementary Figure 5a). However, in Oct4[SMC-P Δ/Δ] corneas, we observed CD31+ tubes devoid of eYFP+ cells at distal regions of the remodeling vasculature, consistent with our earlier observations (Fig. 3b–g). Interestingly, many of the CD31+ EC that lacked eYFP+ cell investment were invested with eYFP−/NG2+ cells (Supplementary Figure 5b). These cells represent a SMC-P population that lacked Myh11 expression and/or failed to undergo recombination at the ROSA locus during the course of tamoxifen injections. This observation suggests that an eYFP− SMC-P population may at least partially compensate for eYFP+ cell dropout following loss of Oct4 in eYFP+ SMC-P.

**SMC-P *Oct4* loss impaired angiogenesis post-HLI**. Next, we sought to determine whether the impaired angiogenesis resulting

from SMC-P *Oct4* knockout extended to other models of angiogenesis. We performed hindlimb ischemia (HLI) surgery, a well-established mouse model of PAD, on Oct4[SMC-P WT/WT] and Oct4[SMC-P Δ/Δ] mice and monitored blood flow recovery over the course of 21 days post-HLI (Fig. 6a), as previously described[26]. There was no significant difference in perfusion recovery as assessed by laser Doppler imaging at days 0 (immediately post-injury), 3, and 7 post-HLI (Fig. 6b, c). At days 0 and 7 post-HLI, there was also no significant difference in capillary density between Oct4[SMC-P WT/WT] and Oct4[SMC-P Δ/Δ] mice (Fig. 6d). However, Oct4[SMC-P Δ/Δ] mice had significantly impaired perfusion recovery at days 14 and 21 post-HLI, compared to Oct4[SMC-P WT/WT] littermate controls (Fig. 6b, c). At day 21 post-HLI, there was also significantly reduced capillary density in SMC-P *Oct4* knockout calf muscle compared to control muscle, consistent with the perfusion deficit detected by Laser Doppler (Fig. 6d, e). There was also a significant decrease in both CD31+ and eYFP+ pixel density, but no change in the ratio of eYFP to CD31, at day 21 post-HLI (Fig. 6f–h). To determine if differences in arteriogenesis, or collateral vessel remodeling, might contribute to the impaired perfusion recovery, we measured arteriogenesis by staining thigh muscle for eYFP and counting the number of vessels with a diameter >10 μm. We did not detect any significant differences in the number of eYFP+ collateral vessels between Oct4[SMC-P WT/WT] and Oct4[SMC-P Δ/Δ] mice (Supplementary Figure 6). Therefore, SMC-P-derived Oct4 plays a major role in angiogenesis, but not arteriogenesis, in mice recovering from hindlimb injury.

Taken together, our results demonstrate that loss of the stem cell pluripotency gene *Oct4* in Myh11-expressing SMC-P results in marked impairment of angiogenesis in two distinct mouse models, corneal alkali burn and HLI.

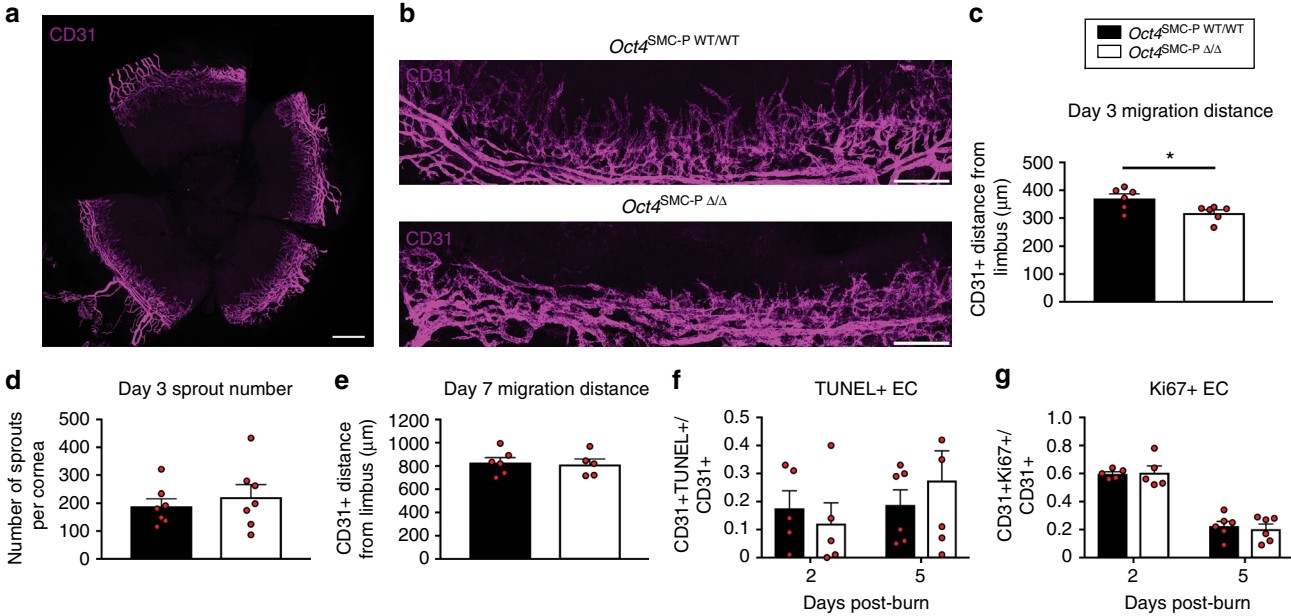

**Fig. 5** SMC-P Oct4 knockout resulted in delayed migration of CD31+ EC following corneal burn. **a** Whole mount immunostaining of a representative Oct4[SMC-P WT/WT] cornea for CD31 (magenta) at day 3 post-corneal burn. Scale bar = 500 μm. **b** Fields of view from Oct4[SMC-P WT/WT] and Oct4[SMC-P Δ/Δ] mice stained with CD31 at day 3 post-corneal burn. Scale bar = 200 μm. **c** Quantification of the average distance of CD31+ EC away from the A–V pair of the corneal limbus at day 3 post-burn. **d** Quantification of EC sprout number throughout the cornea. **e** Quantification of the average distance of CD31+ EC away from the A–V pair of the corneal limbus at day 7 post-burn ((**c–e**) n = 6 WT, 6 KO). **f** Quantification of the number of TUNEL+ EC in cornea at day 2 (n = 5 WT, 5 KO) and day 5 (n = 6 WT, KO) post-burn. **g** Quantification of the number of Ki67+ EC in cornea at day 2 (n = 5 WT, 5 KO) and day 5 post-burn (n = 6 WT, 6 KO). Values = mean ± s.e.m. Statistics were performed using unpaired two-tailed t-test (**c–g**). *P < 0.05

**SMC-P *Oct4* knockout resulted in decreased Slit3 expression.** To identify putative Oct4 target genes, we analyzed previous RNA-Seq data from our lab to determine genes differentially expressed under normoxia versus hypoxia (1% $O_2$) in cultured $Oct4^{WT/WT}$ and $Oct4^{\Delta/\Delta}$ SMC. We used hypoxia as an in vitro stimulus because it is a potent activator of angiogenesis, and we previously showed it activates *Oct4* expression[5]. We then determined which of these hypoxia-regulated genes were differentially expressed in $Oct4^{WT/WT}$ versus $Oct4^{\Delta/\Delta}$ SMC. Finally, we compared the resulting list of genes with putative Oct4 ChIP-seq target genes using the oPOSSUM database. This analysis identified 38 genes that were common in both datasets (Fig. 7a; Supplementary Table 1). Two of these, *Slit3* and *Robo2*, were of particular interest given that SLIT3 is a secreted glycoprotein that has been previously shown to promote migration of multiple cell types including SMC-P[27], EC[27], and macrophages[28] by binding to ROBO receptors on the cell surface[29]. To validate these targets, we measured expression levels in $Oct4^{WT/WT}$ and $Oct4^{\Delta/\Delta}$ cultured SMC by qRT-PCR. Results showed that *Robo2* was decreased in Oct4 knockout SMC, although this did not reach statistical significance (Fig. 7b). *Slit3*, however, was significantly decreased in Oct4 knockout SMC relative to wild type (Fig. 7b). SLIT3 has been shown to promote angiogenesis in multiple in vitro and in vivo systems[27,30–32], suggesting its dysregulation might play a role in our phenotype. To test this, we performed SLIT3 immunostaining following both corneal burn and HLI. Interestingly, SLIT3 protein was expressed in eYFP+ cells of corneal blood vessels measuring ~10–20 μm in diameter (Fig. 7c). In ischemic hindlimb, SLIT3 was expressed by many eYFP+ vessels, including small arterioles, venules, and capillaries (Fig. 7d). SLIT3 levels, normalized to eYFP+ density, were significantly decreased in $Oct4^{SMC-P\ \Delta/\Delta}$ hindlimb compared to $Oct4^{SMC-P\ WT/WT}$ hindlimb following HLI (Fig. 7e), providing further evidence that expression of Slit3 is Oct4-dependent. Additionally, to test the role of SLIT3 on SMC and EC wound

closure, we performed in vitro scratch wound assays, with or without exogenous SLIT3. We found that SLIT3 significantly increased the number of SMC or EC within the scratch wound, supporting a role for SLIT3 in promoting cell migration and/or proliferation in SMC and EC (Fig. 7f–g). Taken together, Oct4-dependent regulation of Slit3 in SMC-P is one potential mechanism by which Oct4 promotes angiogenesis.

## Discussion

In the present study, we identified a role for the stem cell pluripotency gene Oct4 in regulating perivascular cell function following tissue injury or hypoxia-induced angiogenesis. This is only the second report of a functional role for Oct4 in any somatic cell. The first was a previous study by our lab showing that Oct4 expression in SMC is required for formation of a fibrous cap in advanced atherosclerotic lesions. However, the present study is the first to demonstrate a protective role for Oct4 in a process, i.e. angiogenesis, that is critical for survival to and through one's reproductive years. Therefore, a functional role for Oct4 in perivascular cells likely evolved as a means to form new blood vessel networks, and/or repair damaged vessels, and only secondarily proved beneficial during atherosclerosis development.

Of interest, recent studies from our lab demonstrated that Klf4, another stem cell reprogramming factor[33], is also active in SMC-P in a number of processes, including atherosclerosis development, maintenance of resistance arteriole diameter, and arteriole SMC-P coverage[24,34]. Oct4 and Klf4 interact to promote reprogramming to induced pluripotent stem cells (iPSCs)[35]. However, they play distinct roles within SMC-P in the microvasculature. Klf4 serves to stabilize the baseline microvasculature. In contrast, we found no role for Oct4 in baseline microvasculature. Rather, Oct4 serves to stabilize the angiogenic vasculature, as reflected by the significant functional changes we observed upon its loss, including increased leak following corneal burn and impaired

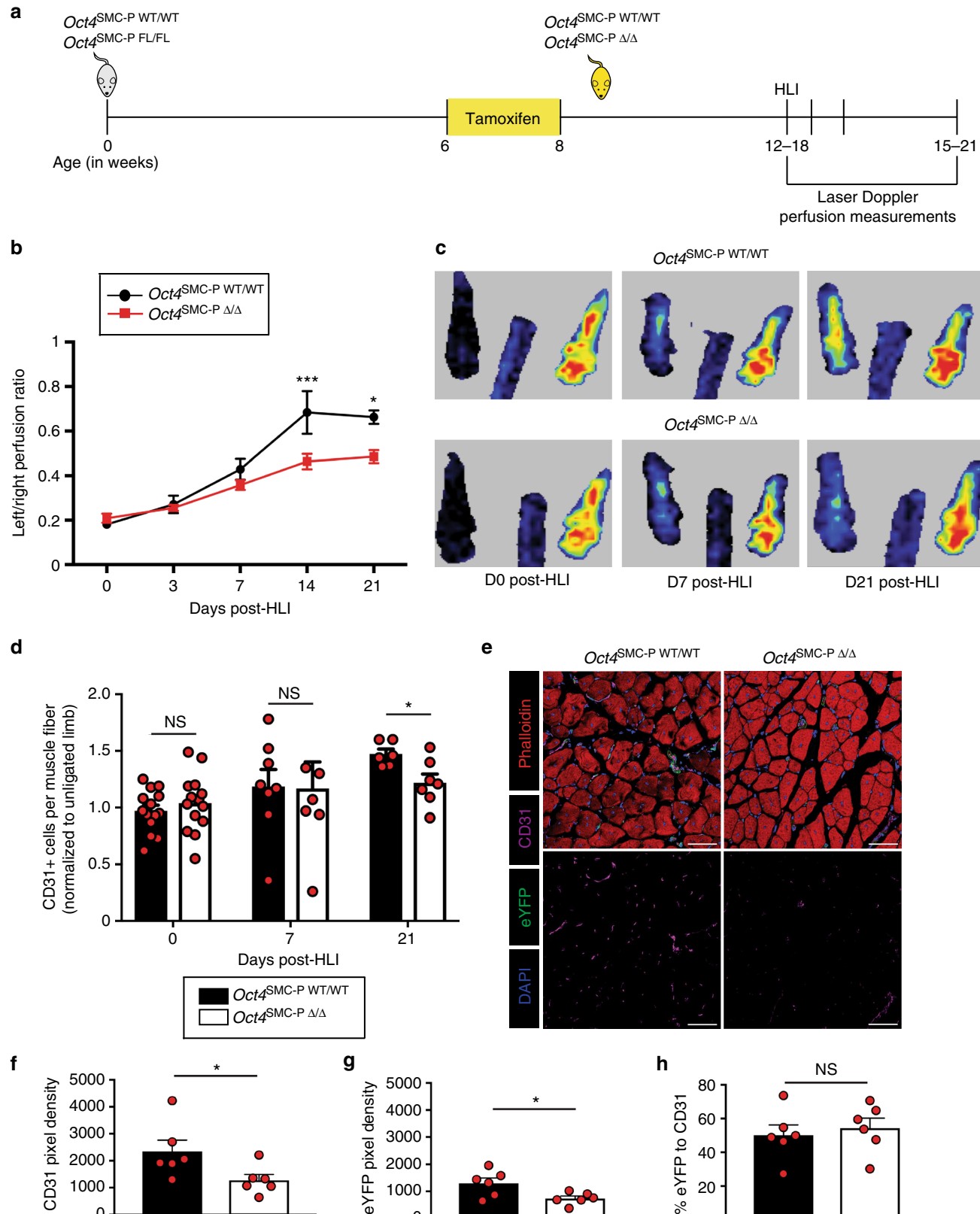

perfusion recovery following HLI. The fact that SMC-P utilize two stem cell pluripotency factors may partially be a reflection of the plasticity of SMC-P, which are non-terminally differentiated cells that can undergo phenotypic modulation to less differentiated states, concomitant with increased proliferative and migratory capacity[36]. For example, roughly half of all eYFP+ cells

in advanced atherosclerotic lesions express markers of other cell types including macrophages, myofibroblasts, and mesenchymal stem cells[24]. Other groups have used lineage tracing to show SMC-P can adopt features of other cell types, such as beige adipocytes following cold stress[37]. However, SMC-P phenotypic switching is not an all-or-none response, and activation of Oct4

**Fig. 6** SMC-P knockout of Oct4 resulted in impaired perfusion recovery and angiogenesis at day 21 post-HLI. **a** Schematic outlining experimental design. **b** Perfusion recovery of Oct4[SMC-P WT/WT] and Oct4[SMC-P Δ/Δ] mice as assessed by laser Doppler perfusion and expressed as left (ligated) over right (unligated) perfusion ratio of the plantar sole ($n = 8$ WT, 9 KO). **c** Representative images of perfusion to the plantar soles at days 0, 7, and 21 post-HLI. **d** Quantification of number of CD31+ cells per muscle fiber at days 0, 7, and 21 post-HLI. For each mouse, data was normalized to the number of CD31+ cells per muscle fiber of the corresponding unligated muscle [$n = 14$ WT, 14 KO (D0); $n = 8$ WT, 6 KO (D7); $n = 6$ WT, 7 KO (D21)]. **e** Representative images of calf muscle cross-sections at day 21 post-HLI stained for DAPI, eYFP, CD31, and Phalloidin. Scale bar = 50 μm. **f**–**g** Quantification of pixel density in calf muscle for CD31 (**f**) or eYFP (**g**) ($n = 6$ WT, 6 KO). **h** Percentage of eYFP to CD31 pixel density as measured in (**f** and **g**) ($n = 6$ WT, 6 KO). Values = mean ± s.e.m. Statistics were performed using two-way ANOVA with Sidak's multiple comparisons test (**b**) and unpaired two-tailed $t$-test (**d**, **f**–**h**). *$P < 0.05$ and ***$P < 0.001$

and/or Klf4 may be required at distinct stages of this process. In the future, it will be intriguing to test whether Oct4, like Klf4, also plays critical functional roles in other cell types, including EC[38,39]. In summary, we have now shown that two stem cell pluripotency factors play distinct protective, rate-limiting roles in SMC-P in the microvasculature, such that conditional deletion of either leads to numerous deleterious downstream changes.

Interestingly, following SMC-P loss of Klf4 in young, healthy non-hyperlipidemic mice, multiple microvascular beds developed gaps in eYFP+ perivascular cell coverage that were filled by Acta2+Myh11+eYFP− cells[34]. Based on bone marrow transfer experiments, we showed that these eYFP− perivascular cells were not of myeloid origin, suggesting an alternative source including: (1) EC underdoing endothelial-to-mesenchymal transition[40], (2) Sca1+ adventitial progenitor cells[41], (3) other SMC-P progenitor populations not labeled by the Myh11− CreER[T2] system, (4) Myh11− SMC-P populations, and/or (5) Myh11+ SMC-P that failed to undergo recombination. In the present study, these same alternative sources of perivascular cells may partially compensate by activating a SMC-P-like program following eYFP+ cell dropout resulting from Oct4 knockout. In other words, when eYFP+ cells labeled by our Myh11−CreER[T2] system lack Oct4 and are unable to appropriately function, alternative sources of perivascular cells may then have a selective advantage in investing nascent EC tubes and partially rescuing the angiogenic phenotype. Indeed, herein we have provided some evidence that this may occur. For example, by crossing our Myh11−CreER[T2] ROSA eYFP lineage tracing mouse to an NG2-dsRED reporter mouse, we found that many of the NG2+ pericytes are eYFP+. However, there is a population of NG2+ SMC-P that lack eYFP expression and as such, retain the ability to activate Oct4 and adequately participate in angiogenesis. In remodeling corneal vasculature of Oct4[SMC-P Δ/Δ] mice, we observed that, despite distal CD31+ tubes being virtually devoid of eYFP+ cells due to loss of Oct4, a population of eYFP−/NG2+ SMC-P were observed surrounding these same distal CD31+ EC tubes. This means that eYFP− SMC-P populations likely partially compensate following loss of the eYFP+ population. Indeed, despite significant reductions in eYFP+ cell density and distribution resulting in increased vascular leak following corneal burn, there were relatively modest effects on EC. Loss of SMC-P coverage has previously been associated with increased EC proliferation[42] and impaired EC sprouting[15]. In contrast, we observed no difference in EC proliferation or sprout number. We did observe a delay in EC migration into the cornea at day 3, but this was completely resolved by day 7. This may be due in part to investment of EC by additional pericyte populations not labeled by our Myh11 lineage tracing system, including a subset of NG2+ pericytes.

However, our data cannot rule out the possibility of eYFP false negatives, or cells that express Myh11 at the time of tamoxifen injection but fail to undergo recombination and therefore are not labeled with eYFP. In Oct4[SMC-P FL/FL] mice, these same cells would

also presumably fail to undergo recombination at the Oct4 locus and would therefore likely have a survival advantage. With recent evidence demonstrating that pathologic remodeling may be due to clonal expansion of a few cells, SMC-P that fail to recombine at the Oct4 locus may then undergo selective expansion[43–45]. Additionally, immunostaining is insufficient to determine cell of origin and can only determine positive or negative marker expression at the time of analysis. Rigorously testing the hypothesis that eYFP− cells compensate following loss of Oct4 in eYFP+ SMC-P would require development of dual conditional lineage tracing models. Importantly, this would require the use of a second, Cre-independent recombinase, such as Dre, since we utilize Myh11-CreER[T2] to generate the Oct4 knockout phenotype. Therefore, fully and rigorously addressing the possibility of compensation by any of the aforementioned candidate cell types would require generating, validating, and repeating experiments in multiple Cre-independent lineage tracing models, coupled with Myh11-CreER[T2] knockout of Oct4.

Activation of the pluripotency factor Oct4 leads to a multitude of downstream changes during its role as a master transcription factor in embryonic stem cells (ESCs)[46]. Interestingly, this is again reflected in adult SMC-P, as our previous in vitro and in vivo RNA-Seq analyses identified thousands of Oct4-dependent genes in SMC[5]. Of major interest, our analyses showed reduced expression of the guidance gene Slit3 following Oct4 knockout both in vitro and in vivo. In vitro experimentation also demonstrated that exogenous Slit3 causes increased SMC and EC wound closure. Therefore, the impaired SMC-P and EC migration in our studies may be due, at least in part, to dysregulated Slit3 signaling in SMC-P. Consistent with these data, numerous studies have previously shown that cell types critical for angiogenesis, including SMC-P[47,48] and EC[49–52], as well as macrophages[28,53] and nerve cells[54], all express ROBO receptors and would therefore be impacted by altered SLIT3 levels in the extracellular space. In the future, it would be interesting to generate SMC-P Slit3 knockout mice to test whether this would recapitulate the phenotype observed in our SMC-P Oct4 knockout mice. However, testing this idea will be extraordinarily difficult for the following reasons: First, there are multiple SLIT ligands, including SLIT1 and SLIT2, that have both overlapping and distinct functions that may compensate for the loss of SLIT3[55]. Second, SLIT3 is produced by multiple cell types in our tissues of interest, including epithelial cells and EC[54] which may upregulate expression in response to altered SLIT3 gradients resulting from decreased production by SMC-P. Third, it is highly unlikely that the phenotype we report here is due exclusively to loss of only one of the many known Oct4-dependent genes. For instance, we also show that Oct4 loss in SMC-P leads to increased cell death one day after corneal burn, similar to the pro-survival role for Oct4 previously described in murine ESCs[56,57]. Furthermore, we have previously shown that numerous other genes previously implicated in control of angiogenesis, including matrix metalloproteinases 3 and 13, multiple collagens, and osteopontin are also Oct4-dependent[5]. Taken together, our results suggest that

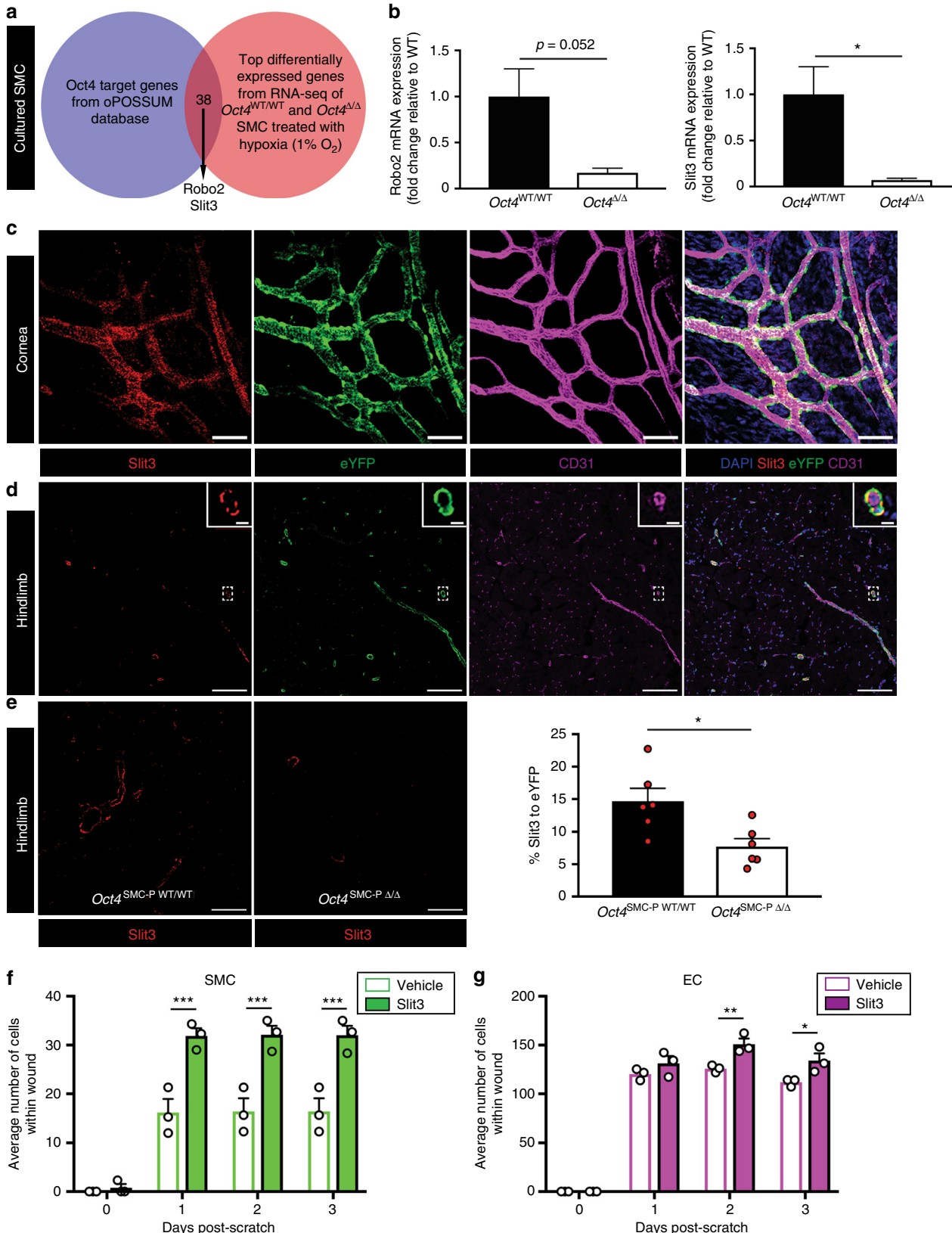

Oct4 in SMC-P leads to numerous downstream changes, including altered Slit3 levels, which in aggregate contribute to the angiogenic phenotype observed in two different in vivo models.

In conclusion, this study provides multiple lines of evidence that SMC-P Oct4 is critical for angiogenesis following either corneal alkali burn or HLI. Results are of major interest and

significance since they indicate that the reactivation of Oct4, which we previously demonstrated occurs in SMC during development of atherosclerotic lesions, is not exclusive to that pathological state[5]. Rather, we show that Oct4 reactivation within perivascular cells is required for functional network formation in response to tissue injury or hypoxia. Moreover, whereas Oct4 has

**Fig. 7** Slit3 was expressed in eYFP+ cells and was decreased following Oct4 knockout. **a** We found 38 genes that were both differentially expressed in our RNA-seq analysis of cultured Oct4$^{WT/WT}$ and Oct4$^{\Delta/\Delta}$ SMC treated with hypoxia (1% O$_2$), as well as putative Oct4 target genes using the oPOSSUM database. Two of these genes (*Robo2* and *Slit3*) were members of the Slit-Robo pathway of guidance genes. **b** qRT-PCR of Oct4$^{WT/WT}$ and Oct4$^{\Delta/\Delta}$ cultured SMC for expression of *Robo2* and *Slit3*. Data is expressed as fold change relative to wild type. Data is from three independent experiments. **c** Immunostaining of whole mount cornea at day 3 post-corneal burn for CD31, eYFP, and SLIT3. Scale bar = 50 μm. **d** Immunostaining of ischemic calf muscle at day 7 post-HLI for DAPI, CD31, eYFP, and SLIT3. Scale bar = 100 μm. Zoom in scale bar = 10 μm. **e** Representative images and quantification of SLIT3 pixel density, normalized to eYFP+ pixel density, in ischemic calf muscle at day 21 post-HLI. Scale bar = 50 μm (n = 6 WT, 6 KO). **f–g** Scratch wound assays showing number of SMC (**f**) or EC (**g**) within the scratch wound following vehicle or Slit3 (1 nmol l$^{-1}$) treatment. Values = mean ± s.e.m. Statistics were performed using unpaired two-tailed *t*-test (**e**) or two-way ANOVA with Sidak's multiple comparisons test (**f–g**). *P < 0.05, **P < 0.01, ***P < 0.001

been shown to be essential for maintenance of the pluripotency state of embryonic stem cells, herein we describe a role for this factor in regulating cell survival, cell migration, investment of endothelial tubes, and vessel permeability. Further studies will be required to better understand the mechanisms regulating these effects, as well as to determine how this information might be exploited to either inhibit or promote effective angiogenesis in different disease states.

## Methods

**Mice**. To generate Myh11-CreER$^{T2}$ ROSA floxed STOP eYFP Oct4$^{FL/WT}$ mice, we first bred *Oct4*$^{FL/FL}$ (Pou5f1$^{tm1Scho}$)[4] mice with Myh11-CreER$^{T2}$ (Tg(Myh11-cre/ERT2)1Soff)[58] mice. This yielded *Oct4*$^{FL/WT}$ Myh11-CreER$^{T2}$ mice. The Myh11-CreER$^{T2}$ transgene is on the Y chromosome, precluding the use of Cre-negative mice as controls. We then bred Myh11-CreER$^{T2}$ Oct4$^{FL/WT}$ males with *Oct4*$^{FL/WT}$ females to generate Myh11-CreER$^{T2}$ *Oct4*$^{FL/FL}$ and Myh11-CreER$^{T2}$ *Oct4*$^{WT/WT}$ male littermates. We then crossed ROSA26-STOP$^{flox}$YFP$^{+/+}$ mice (B6.129×1-Gt (ROSA)26Sortm1(eYFP);Cos/J) with Myh11-CreER$^{T2}$ mice to yield Myh11-CreER$^{T2}$ ROSA26-STOP$^{flox}$YFP$^{+/+}$ mice using the same strategy described above. We then crossed *Oct4*$^{FL/FL}$;Myh11-CreER$^{T2}$ male mice with Myh11-CreERT2 ROSA26-STOP$^{flox}$YFP$^{+/+}$ mice to generate *Oct4*$^{FL/WT}$ Myh11-CreER$^{T2}$ ROSA floxed STOP eYFP males and *Oct4*$^{FL/WT}$ ROSA floxed STOP eYFP female mice. The resulting mice were then backcrossed nine generations to the C57BL/6 line (Jackson Labs; Bar Harbor, ME). We then used these mice as breeders to generate Myh11-CreER$^{T2}$ ROSA floxed STOP eYFP Oct4$^{WT/WT}$ and Myh11-CreER$^{T2}$ ROSA floxed STOP eYFP Oct4$^{FL/FL}$ male littermate mice which were genetically identical other than containing WT versus floxed Oct4 alleles. We achieved Cre-mediated recombination via 10 daily intraperitoneal injections of tamoxifen (Sigma-Aldrich; St. Louis, MO) (1 mg in 100 μl of peanut oil (Sigma-Aldrich; St. Louis, MO)) between 6 and 8 weeks of age. All experimental mice received identical amounts of tamoxifen. We only used male progeny for experiments, since the Myh11-CreER$^{T2}$ transgene is located on the Y chromosome. Upon tamoxifen treatment, there is simultaneous activation of eYFP and excision of the floxed Oct4 alleles, generating Myh11-CreER$^{T2}$ ROSA eYFP Oct4$^{WT/WT}$ and Myh11-CreER$^{T2}$ ROSA eYFP Oct4$^{\Delta/\Delta}$ mice. Henceforth, we refer to them as Oct4$^{SMC-P\ WT/WT}$ and Oct4$^{SMC-P\ \Delta/\Delta}$, for simplicity. At 12–18 weeks of age, mice were used for experiments. We also crossed our Myh11-CreER$^{T2}$ ROSA floxed STOP eYFP mice with NG2-DsRED (Tg(Cspg4-DsRed.T1)1Akik/J) (Jackson Labs; Bar Harbor, ME) mice to generate NG2-DsRED Myh11-CreER$^{T2}$ ROSA floxed STOP eYFP mice. We treated these mice with the same tamoxifen protocol outlined above to generate NG2-DsRED Myh11-CreER$^{T2}$ ROSA eYFP mice. Protocols for studies involving mice were approved by the University of Virginia Animal Care and Use Committee. Studies complied with all relevant ethical regulations.

**Whole mount retina imaging**. Thirty minutes prior to sacrifice, isolectin (IB4-Alexa647; Life Technologies; Carlsbad, CA) was injected into the mouse tail vein. Mice were euthanized by CO$_2$ asphyxiation and then whole retinas were removed and flat-mounted on slides to image isolectin plus endogenous NG2-dsRED and eYFP fluorescence. Images were taken on a Zeiss LSM700 scanning confocal microscope on ×20 magnification.

**eYFP+ cell sorting**. Mice were euthanized with CO$_2$ asphyxiation and perfused with 10 ml PBS. Tissues were harvested and digested with 4 U ml$^{-1}$ Liberase TM (Roche 05401119001) and 0.74 U ml$^{-1}$ Elastase in RPMI-1640 for 1.5 h at 37 °C. Tissues were spun down at 1000×g for 10 min, resuspended in FACS buffer and run through a 70 μm cell strainer. Samples were run on a Becton Dickinson FACSVantage SE Turbo Sorter and sorted based on native eYFP expression.

**Blood pressure telemetry**. Blood pressure and heart rate were measured using radiotelemetry units (Data Sciences International; St. Paul, MN). The catheter of a radiotelemetry unit was inserted into the left carotid artery and the radio-transmitter was placed in a subcutaneous pouch on the right flank. Blood pressure

was recorded for 5 min per hour for 2 days on, then 5 days off, for a total of 10 days of recording. Recordings were limited to weekend days to minimize noise and stress-related fluctuations in blood pressure.

**Corneal alkali burn**. At 12–18 weeks of age, mice were anesthetized with iso-fluorane (2% isoflurane, 200 ml min$^{-1}$ flow rate). A drop of sterile 0.5% pro-paracaine hydrochloride ophthalmic solution was added as a topical anesthetic to numb the eye 2 min prior to burn and another drop applied immediately prior to burn. Corneal alkali burn was induced by applying applicator sticks coated with 75% AgNO$_3$/25% KNO$_3$ (SnypStix by Grafco; Atlanta, GA) to the center of the right cornea for 10 s[25]. An additional drop of proparacaine was applied to the right cornea immediately post-burn, and mice received post-operative analgesic (buprenorphine 0.1–0.2 mg kg$^{-1}$ subcutaneous). For each experimental mouse, the contralateral eye remained unburned and untreated.

**Intravital confocal microscopy and quantification**. Animals were anesthetized with an intraperitoneal injection of ketamine/xylazine/atropine (60/4/0.2 mg kg$^{-1}$ body weight) (Zoetis; Kalamazoo, MI/West-Ward; Eatontown, NJ/Lloyd Laboratories; Shenandoah, IA). A drop of sterile 0.5% proparacaine hydrochloride oph-thalmic solution was added as a topical anesthetic to numb the eye before imaging. Ophthalmic lubricant Genteal gel (Alcon; Fort Worth, TX) was applied to the eye during imaging to prevent drying. Mice were placed on a microscope stage that contained a warming pad to maintain a constant body temperature of 37 °C, eye-lashes and whiskers were gently pushed back with ophthalmic lubricant Genteal gel, and the snout was gently restrained with a nosecone. Mice were imaged immedi-ately (within 10 min) prior to injury (day 0), and then at days 3, 7, and 21 post-injury on a confocal microscope (Nikon Instruments Incorporated, Melville, NY; Model TE200-E2; ×10, ×20, and ×60 objectives optimized for three channels-laser excitation wavelengths at 488, 543, and 632). Immediately prior to each round of imaging, mice were perfused with isolectin GS-IB4 (lectin) or rhodamine-dextran (MW 70 kDa, Sigma-Aldrich) via retro-orbital injection into the contralateral eye to visualize perfusion-competent vasculature. During imaging, the right eye was placed against a coverslip that rested on the stage of the inverted confocal microscope. The entire portion of the eye that rested on the stage, or approximately one quarter of the entire corneal circumference, was imaged on ×20 or ×60 magnification by taking multiple fields of view (FOV) using Z-stack imaging through the entire cornea thickness. Each FOV was compressed into a maximum intensity projection[25]. All FOV from each eye were stitched together into montages using Adobe Illustrator and then analyzed. To rigorously quantify the number of eYFP+ cells within each montage, the corneal vascular networks were divided into distinct regions. The number of cells on the arteriole–venule (A–V) pair was counted to determine the number of eYFP+ cells per A–V pair. To quantify number of eYFP+ cells beyond the A–V pair in the direction of the center of the cornea, we divided the vascular network into 50 μm regions, where 50 μm was measured from the A–V pair towards the cornea center along the entire length of the montaged image (Supplementary Figure 3). The number of eYFP+ cells in each of these regions was counted and normalized to the area of the respective region. Researchers were blinded to the genotype of the animals until the end of analysis.

**Bright field imaging and quantification of hemorrhaging**. Bright field images of corneas under ×4 magnification were obtained using a Nikon Digital Sight DS-L2 Camera Controller (Nikon Instruments Inc., Melville, NY; Model 214602) to assess the network-wide hierarchy of neovessels and determine the macrostructural health of the tissue (Table 1). For each mouse, multiple fields of view were taken encompassing the entire circumference of the eye. Corneal hemorrhaging in bright field montages was graded on a scaled score from 0 to 7, adapted from Kisucka et al.[59].

**Rhodamine dextran injections**. Mice were imaged at day 3 post corneal-burn. Following anesthetization with ketamine/xylazine/atropine, mice were adminis-tered a retro-orbital injection of 70 kDa rhodamine–dextran (Sigma-Aldrich) immediately prior to imaging, such that movie recording started <3 min post-injection. Consequently, the perfusion and leak of dextran have just begun at the onset of recording. Digital images of the vascular networks were acquired using a NikonTE200-E2 confocal microscope, as described above. One field of view per

cornea was imaged with full-thickness z-stacks (25–30 slices at 3 µm between each slice) on repetition for 90 min. Volume renders of z-stacks, using the maximum intensity projection, were used to capture the entire corneal vascular network in the field of view. Movie files were analyzed in ImageJ, where we measured the mean pixel intensity in three equal-size regions of interest (ROIs) that were evenly distributed across three different areas in the field of view (above limbus, in vascular loops within the limbus, and below the A–V pair defining the start of the limbus), for a total of nine ROIs analyzed per frame. The three ROIs within a given area were then averaged for each frame and recorded as a single value. These values were then plotted against time. Finally, we quantified the area under the curve to capture the total leak of dextran from the mouse over time. Researchers were blinded to the genotype of the animals until the end of analysis.

**Whole mount cornea immunostaining.** Mice were euthanized by $CO_2$ asphyxiation and then whole corneas, including the limbus, were removed and washed in phosphate-buffered saline (PBS; Life Technologies; Carlsbad, CA). Corneas were then fixed in 1% paraformaldehyde (PFA; Electron Microscopy Sciences; Hatfield, PA) for 45 min, followed by blocking for 1 h in blocking buffer containing 10% horse serum. Primary anti-GFP antibody (Abcam ab6673, 1:100) was added overnight at 4 °C in blocking buffer. Corneas were then washed and stained for 1.5 h in blocking buffer with CD31 (Dianova 310, 1:250) and/or Slit3 (Sigma SAB2104337, 1:50). Corneas were then washed in PBS+0.1% Tween-20, followed by staining with appropriate secondary antibodies for 1.5 h in blocking buffer. Secondary antibodies used include Donkey anti-goat 488 (Invitrogen, A11055, 1:250), Donkey anti-rabbit 555 (Invitrogen, A31572, 1:250), and Donkey anti-rat 650 (Abcam, ab102263, 1:250). Corneas were washed in PBS+0.1% Tween-20, stained with DAPI (1:100) for 10 min, washed with PBS, and then flat-mounted on slides using 50/50 PBS/glycerol. Images were acquired at ×10 magnification on a Zeiss LSM700 scanning confocal microscope using full-thickness Z stacks (~15–20 slices at 5 µm between each slice). For each cornea a minimum of 2 FOV from different leaflets were used for analysis. Individual slices were collapsed into maximum intensity projections for analysis using Zen 2009 Light Edition Software or Image-Pro Plus. Researchers were blinded to the genotype of the animals until the end of analysis.

**Harvesting, immunostaining, and analysis of tissue.** Mice were euthanized by $CO_2$ asphyxiation and then perfused through the left ventricle with 5 ml of PBS, followed by 10 ml of 4% periodate–lysine–paraformaldehyde (PLP), followed by 5 ml of PBS. Whole eyes were fixed for an additional 2 h in 4% PLP. Tissues were then processed through a sucrose gradient and embedded in optimal cutting temperature (OCT) compound, frozen in liquid nitrogen, and stored at −80 °C until sectioning. Eyes, calf muscles, and/or thigh muscles were serially sectioned at 5–10 µm thickness on a cryostat. For immunohistochemical staining, slides were blocked for 1 h in blocking buffer containing 10% horse serum + 0.6% fish skin gelatin in PBS. Primary antibodies were added overnight in blocking buffer at 4 °C, and then tissues were washed ×2 with PBS-Tween (0.1%). Secondary antibodies were added for 1 h at room temperature in blocking buffer. Tissues were then washed ×2 with PBS-Tween (0.1%), counterstained with DAPI, and cover slipped using Prolong Gold mounting medium. Slides were stained using combinations of the following primary antibodies: GFP (Abcam ab6673, 1:250), CD31 (Dianova 310; 1:250), Ki67 (abcam ab15583, 1:100), NG2 chondroitin sulfate proteoglycan (Millipore AB5320, 1:500), PDGFRβ (abcam ab32570, 1:500), and Slit3 (Sigma SAB2104337, 1:50). Secondary antibodies used for immunostaining include Donkey anti-goat 488 (Invitrogen, A11055), Donkey anti-goat 555 (Invitrogen, A21432), Donkey anti-goat 647 (Invitrogen, A21447), Donkey anti-rabbit 488 (A21206), Donkey anti-rabbit 555 (Invitrogen, A31572), Donkey anti-rat 550 (Abcam, ab102261), and Donkey anti-rat 650 (Abcam, ab102263). All secondary antibodies were used at 1:250 concentration. Isotype-matched IgG antibodies were used as a negative control for all antibodies. For TUNEL staining, dUTP conjugated to CF 640R dye (Biotium, 1:50) was used. DAPI (1:100) was used to label nuclei and Alexafluor 555 Phalloidin (1:1000) was used to label muscle fibers. No TdT enzyme was used as a negative control for TUNEL staining. Images were acquired using a Zeiss LSM700 scanning confocal microscope. A minimum of 3 FOV from multiple locations throughout the tissue were analyzed. High-resolution Z-stack analysis was performed using Zen 2009 Light Edition Software to ensure staining was limited to a single cell. For analysis of eye cross-sections, quantification was limited to cells comprising the corneal vasculature. For pixel analysis, Z-stack slices were collapsed into maximum intensity projections and analyzed using Image-Pro Plus. Researchers were blinded to the genotype of the animals until the end of analysis.

**HLI model.** At 12–18 weeks of age, mice were anesthetized with ketamine (90 mg kg$^{-1}$) and xylazine (10 mg kg$^{-1}$) and then subjected to unilateral femoral artery ligation and resection[26]. Blood flow in the plantar soles was monitored with a laser Doppler perfusion imaging system (Perimed, Inc., Ardmore, PA) immediately after surgery (day 0) and then at days 3, 7, 14, and 21 post-surgery. Mice were placed on a warming pad during surgery and during laser Doppler image acquisition to maintain a constant body temperature of 37 °C. Perfusion was expressed as the ratio of the left (ischemic) to right (nonischemic) hindlimb. The right hindlimb served as an internal control for

each mouse. Animals received buprenorphine (intraperitoneal) immediately after surgery and every 8–12 h thereafter until 48 h post-surgery.

**RNA-Seq analysis.** Genes differentially expressed in hypoxic with respect to normoxic conditions were found for both Oct4 knockout (comparison 1) and wild-type (comparison 2) cultured mouse aortic SMC. Genes differentially expressed in comparison 1 with respect to comparison 2 were found. Since this comparison yielded a high number of genes, strict cuts of 5 in the absolute value of $log_2$ fold change and 0.01 in false discovery rate were applied. oPOSSUM web-based tool[60–62] was used to find transcription factors targeting the resulting list of genes (193 genes after the cuts). The tool was run with the following parameters: all 29,347 oPOSSUM database genes used as a background; all vertebrate profiles with a minimum specificity of 8 bits used; Transcription factor binding site search parameters: conservation cutoff of 0.3, matrix score threshold of 85%, amount of upstream/downstream sequence of 5000/5000. Ninety-six transcription factors, including Oct4, targeting the top genes with $|Z| > 2$ were found.

**Cell culture.** $Oct4^{+/+}$ and $Oct4^{\Delta/\Delta}$ mouse aortic SMC were isolated from thoracic aortas of 6-week-old $Oct4^{SMC-P\ WT/WT}$ and $Oct4^{SMC-P\ \Delta/\Delta}$ mice after tamoxifen injections[5]. Perivascular fat was removed, and aortas were digested in 1 mg ml$^{-1}$ collagenase II, 0.744 U ml$^{-1}$ Soybean Trypsin inhibitor (all Worthington Biochemical Corp.) in Hank's balanced salt solution for 10 min. The adventitia was carefully removed, and the intimal surface was scraped to remove EC. Aortas were cut into ~0.5 mm pieces and placed in the enzyme solution for another 1–1.2 h. Disaggregated SMC were maintained in 20% serum-containing media (DMEM/F12 (Gibco), Fetal bovine serum (FBS; Hyclone), 100 U ml$^{-1}$ penicillin/streptomycin (Gibco), 1.6 mM l$^{-1}$ L-glutamine (Gibco)). After two passages, SMC were switched to 10% serum-containing media. Mouse aortic EC were purchased from Cell Biologics (C57-6052) and maintained in complete mouse endothelial cell medium plus kit (M1168). For experiments, cells were grown to 80–100% confluency and then switched to serum-free media (SFM) for at least 24 h prior to harvest and/or treatment.

**RNA isolation, cDNA preparation, and qRT-PCR.** Total RNA was harvested using phenol–chloroform extraction (TRIzol, Life Technologies, Grand Island, NY). 0.5–1 µg of RNA was reverse-transcribed with iScript cDNA syntesis kit (Bio-Rad). Real-time qRT-PCR was performed on a C1000 Thermal Cycler CFX96 (Bio-Rad) using SensiFAST SYBR NO-ROX mix (Bioline) and primers specific for mouse *Robo2, Slit3,* and *B2M.* Expression of genes was normalized to *B2M.*
*Robo2* For: CGAGCTCCTCCACAGTTTGT
Rev: GTAGGTTCTGGCTGCCTTCT
*Slit3* For: AGTTGTCTGCCTTCCGACAG
Rev: TTTCCATGGAGGGTCAGCAC
*B2M* For: ATGGCTCGCTCGGTGACCCT
Rev: TTCTCCGGTGGGTGGCGTGA

**Scratch wound assay.** Scratch wound assays were performed in six-well plates. Once cells were 100% confluent, cells were washed ×2 with 1×PBS and placed in serum-free media (SFM) for 24 h. A scratch down the middle of each well was made using a p200 pipet tip and then media was immediately replaced with SFM containing 1 nmol l$^{-1}$ SLIT3 (MyBioSource.com, MBS2010323) or Vehicle (PBS). Images were taken immediately after scratch and every 24 h thereafter for 72 h post-scratch on ×4 magnification using an inverted microscope. Prior to cell plating, horizontal lines were made on the underside of each well at 12, 20, and 28 mm from the bottom of each well to ensure the same FOV was imaged at each time point. Numbers of cells that migrated into the scratch were counted in each FOV using ImageJ analysis software. Counts were then averaged together across 9 FOV per condition (3 wells with 3 FOV each).

**Statistical analysis.** Statistical analysis was performed using GraphPad Prism Version 7 software. Normality of the data was determined using a Kolmogorov–Smirnov test. For comparison of two groups with normal distribution, an unpaired two-tailed t-test was used. For comparison of two groups with non-normal distribution, a Mann–Whitney U-test was used. For analysis of three or more groups with normal distribution, a two-way ANOVA was used. All results are presented as mean ± SEM. $P < 0.05$ was considered significant. For all in vivo experiments, each 'n' refers to a biologically independent animal. For all in vitro experiments, each experiment was performed in triplicate and repeated independently three times. Specific statistical tests and number of mice used for each in vivo analysis are reported in the figure legends.

**Reporting Summary.** Further information on experimental design is available in the Nature Research Reporting Summary linked to this Article.

## Data availability

All data that supports the results of this study are available from the corresponding author upon reasonable request. The GEO accession number for the RNA-Seq data set is GSE75044.

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

## Acknowledgements

We would like to thank all members of the Owens lab for their input. We acknowledge Melissa Bevard, Coral Kasden, Angela Washington, and Mary McCanna for help with histology. We acknowledge Liming Yu and Rupa Tripathi for help with cell culture experiments. We acknowledge Gabriel Alencar for analysis of telemetry data. This work was supported by American Heart Association (AHA) Predoctoral Fellowship 15PRE25670040 (to D.L.H.), AHA Predoctoral Fellowship 16PRE3097006 (to M.R.K-G.), AHA Innovative Research Grant 17IRG33370017 (to O.A.C.), NIH Grants R01 HL082838, R01 Ey022063, and The Hartwell Foundation (to S.M.P.), NIH Grants 1R01 HL12635, 1R01 HL116455, 2R01 HL101200 (to B.H.A.), and NIH Grants R01 HL057353, R01 HL135018, and T32 HL007284 (to G.K.O.). This work was also supported by the Wagner Fellowship (M.R.K.-G.).

## Author contributions

D.L.H. and M.R.K.-G. designed and performed experiments, analyzed data, and interpreted data. D.L.H. prepared the figures and wrote the manuscript. M.R.K.-G. contributed to writing and played a major role in editing of the manuscript. O.A.C. provided guidance throughout the project and assisted with data analysis. A.T.N. performed telemetry surgeries and retina staining. R.A.B. performed recombination analysis. S.T. performed RNA-Seq analysis. B.H.A. and S.M.P. assisted with experimental design and data interpretation. G.K.O. supervised the project and had a major role in experimental design, data interpretation, and writing the manuscript. All authors played a role in discussions as well as final editing and approval of the manuscript.

## Additional information

**Competing interests:** The authors declare no competing interests.

