## [Peer Review File · Nature Communications]

Reviewers' comments:

Reviewer #1 (Remarks to the Author):

Hess et al., address the role of perivascular cells in pathological angiogenesis in the corneal alkali burn and hindlimb ischemia (HLI) models. They convincingly show that deletion of Oct4 expression in perivascular cells using Oct4^{fl/fl}/Myh11-CreERT2 mice resulted in decreased mural cell density and blood vessel growth in both injury models. Changes in perivascular cell coverage increased vascular leakage and hemorrhage following corneal alkali burn. The authors show that inhibition of Oct4 expression decreased expression of Slit3 and its receptor Robo2 in vivo and in vitro. While this is not followed up on beyond expression studies, the authors make a convincing argument that a) comprehensive analysis of Slit-Robo signaling will require extensive in vivo work, and b) other factors in addition to Slit/Robo may contribute to the Oct 4 phenotype. Overall, this study identifies SMC-P Oct4 as critical for SMC-P migration and pathological angiogenesis. The data are solid and the manuscript is well written and easy to follow, and identifies reactivation of the pluripotency gene Oct4 as a novel regulator of vascular injury responses in various settings in addition to atherosclerosis.

Reviewer #2 (Remarks to the Author):

This interesting report by Dr Owens et al presents multiple lines of evidence demonstrating a role for perivascular cell-specific expression of Oct4 in aberrant angiogenesis. The strengths of the manuscript include the use of conditional transgenesis for analyses with high temporal and spatial resolution as well as studies including multiple target organs. A minor weakness is the rather lengthy an unfocused introduction which perhaps belongs to a review article. The work is certainly impactful for different fields considering the pleiotropic effects of adult angiogenesis.

Reviewer #3 (Remarks to the Author):

Previous work from the Owens group has suggested that reactivation of the pluripotency factor Oct4 plays a role in mural cell plasticity using a mouse model of atherosclerosis (Cherepanova 2016, Nat Med). The group also demonstrated that Oct4 deletion results in reduced SMC migration in vitro, which was accompanied by reduced expression of matrix metalloproteinases (Cherepanova 2016, Nat Med).

Here, the group tests the hypothesis that Oct4 reactivation also plays a role in physiologically relevant processes. This is tested by combining a floxed Oct4 allele with a Myh11-driven inducible Cre, which result in tamoxifen-induced expression of a recombination reporter (eYFP) in 80% of pericytes (P; Ng2- or Pdgfra-positive) in addition to previously demonstrated smooth muscle cells (SMC). The target cell population is referred to as SMC/P. The authors convincingly demonstrate a phenotype of SMC/P specific Oct4 ablation in two widely used angiogenesis models, with reduced reperfusion after hindlimb ischemia (femoral artery ligation) and increased vascular leakage after corneal burn in Oct4 mutants compared to controls. Group sizes, quantification and statistical analysis are all appropriate and the methods are clearly described.

While SMC/P-specific loss of Oct4 does not affect vascular structure in healthy animals, the number of reporter-expressing cells (used as a proxy for Oct4 deletion, see point 6 below) was significantly reduced after injury in both models. This is relatively unexplored (point 1 below). The authors conclude that the defective angiogenesis results from reduced migration in Oct4-mutant SMC/P. However, convincing evidence for a migration defect is not presented (see point 2 below). Analysis of RNAseq data from in vitro cultured cells exposed to hypoxia suggests that Oct4 directly regulates Slit3/Robo2 in SMC/Ps, but this is not functionally tested (see point 3 below).

This work is an important demonstration that Oct4 reactivation in somatic cells plays a role in angiogenesis. However, insight into the cellular phenotype and/or molecular mechanism would increase the general interest in this observation.

Major points

1. The reduced number of labelled cells after injury suggests that Oct4 deletion either decreased cell survival or proliferation (or both). This apparent hypersensitivity to damage/hypoxia is an important finding, which may explain the observed defects in angiogenesis. It would be informative to know whether this is due to increased apoptosis and/or reduced proliferation at early timepoints after injury. In the corneal burn model, apoptosis is scored at day 2 and proliferation at day 5, whereas the reduced cell number is evident at all timepoints analysed (from day 3).
2. In the corneal burn model, the number of eYFP+ cells is reduced at all timepoints and all distances in Oct4-mutants relative to controls. It is therefore not surprising that the number of labelled cells that have migrated is also reduced. The conclusion that migration is affected should either be toned down or substantiated for example with cell biology evidence (for example changes in cytoskeletal arrangement) or elaboration of the Slit3 phenotype (see point 3).

3. The functional relevance of the proposed Oct4 target Slit3 is not at all evident. Firstly, the verification of reduced levels of Slit3 in vivo is not convincing. Since there is a large reduction in the number of recombined (eYFP+) cells in Oct4-mutant animals and Slit3 is expressed in SMC/P's it is not surprising that Slit3 "pixel intensity" is reduced (Figure 7e). Quantification should be limited to eYFP+ cells (if this is already the case, it should be clearly stated in the legend). Secondly, no evidence that Slit3 is a target of Oct4 or that its expression is sensitive to hypoxia in SMC/Ps is provided. Finally, it is not clear that reduced levels of Slit3 is even related to the observed angiogenesis defect. This could be examined in vitro or by virus-mediated overexpression in vivo and could potentially strengthen the claims that this is a potential mechanism of impaired migration.

Alternatively to analysis of Slit3 functional relevance, more insight into the cellular phenotype and Oct4 targets could be gained from transcriptomic profiling of eYFP+ cells isolated from animals after injury.

4. As discussed, the reduced number of labelled cells in Oct4-mutant animals could result in compensatory activation of progenitor cells in the vascular wall. It would be important to assess the extent of non-labelled cells acquiring a SMC/P phenotype, for example by staining for pericyte markers (Ng2).

Minor points

5. It is not clear whether the baseline quantification of eYFP+ cells shown in Figure 2e was performed immediately before injury. If not, quantification immediately before injury should be included to eliminate the possibility that Oct4 deletion affects cell survival or turnover.

6. Efficient Oct4 deletion in reporter-expressing cells is not verified. Recombination efficiency is cell type and locus dependent (Vooijs and Berns, 10.1093/embo-reports/kve064) and it is important to check that reporter expression reflects deletion of both Oct4 alleles at least at the genomic level.

7. It appears that dextran leakage has already occurred at the start of the supplementary videos. Please specify when the recordings were made relative to the injection.

8. Individual data points should be indicated in bar graphs (journal policy).

Reviewer #1 (Remarks to the Author):

We thank the reviewer for their thoughtful review and comments regarding our manuscript. For ease of review, the reviewer's comments are indicated in bold, followed by our response below.

Hess et al., address the role of perivascular cells in pathological angiogenesis in the corneal alkali burn and hindlimb ischemia (HLI) models. They convincingly show that deletion of Oct4 expression in perivascular cells using Oct4^{fl/fl}Myh11-CreERT2 mice resulted in decreased mural cell density and blood vessel growth in both injury models. Changes in perivascular cell coverage increased vascular leakage and hemorrhage following corneal alkali burn. The authors show that inhibition of Oct4 expression decreased expression of Slit3 and its receptor Robo2 in vivo and in vitro. While this is not followed up on beyond expression studies, the authors make a convincing argument that a) comprehensive analysis of Slit-Robo signaling will require extensive in vivo work, and b) other factors in addition to Slit/Robo may contribute to the Oct 4 phenotype. Overall, this study identifies SMC-P Oct4 as critical for SMC-P migration and pathological angiogenesis. The data are solid and the manuscript is well written and easy to follow, and identifies reactivation of the pluripotency gene Oct4 as a novel regulator of vascular injury responses in various settings in addition to atherosclerosis.

The authors thank the reviewer for the kind remarks and for appreciating the extreme complexity and difficulty of *in vivo* mechanistic studies of the Slit/Robo pathway.

Reviewer #2 (Remarks to the Author):

We thank the reviewer for their thoughtful review and comments regarding our manuscript. For ease of review, the reviewer's comments are indicated in bold, followed by our response below.

This interesting report by Dr Owens et al presents multiple lines of evidence demonstrating a role for perivascular cell-specific expression of Oct4 in aberrant angiogenesis. The strengths of the manuscript include the use of conditional transgenesis for analyses with high temporal and spatial resolution as well as studies including multiple target organs. A minor weakness is the rather lengthy and unfocused introduction which perhaps belongs to a review article. The work is certainly impactful for different fields considering the pleiotrophic effects of adult angiogenesis.

We thank the reviewer for the positive comments. The Introduction was written with the broad, diverse readership of Nature Communications in mind. Therefore, we feel the information in the Introduction is necessary to fully appreciate the limitations in our field that this study intended to address. Of course, we will shorten the Introduction if the Editor feels this is required.

Reviewer #3 (Remarks to the Author):

We thank the Reviewer for the extremely high-quality review, diligent reading, and thorough analysis of our manuscript. We have attempted to address each of the reviewer's concerns and are excited to report new data inspired by the reviewer's helpful comments. Indeed, we believe the paper is much improved from the original submission. For ease of review, the reviewer's comments are indicated in bold, followed by our response below.

Previous work from the Owens group has suggested that reactivation of the pluripotency factor Oct4 plays a role in mural cell plasticity using a mouse model of atherosclerosis (Cherepanova 2016, Nat Med). The group also demonstrated that Oct4 deletion results in reduced SMC migration in vitro, which was accompanied by reduced expression of matrix metalloproteinases (Cherepanova 2016, Nat Med).

Here, the group tests the hypothesis that Oct4 reactivation also plays a role in physiologically relevant processes. This is tested by combining a floxed Oct4 allele with a Myh11-driven inducible Cre, which results in tamoxifen-induced expression of a recombination reporter (eYFP) in 80% of pericytes (P; Ng2- or Pdgfra-positive) in addition to previously demonstrated smooth muscle cells (SMC). The target cell population is referred to as SMC/P. The authors convincingly demonstrate a phenotype of SMC/P specific Oct4 ablation in two widely used angiogenesis models, with reduced reperfusion after hindlimb ischemia (femoral artery ligation) and increased vascular leakage after corneal burn in Oct4 mutants compared to controls. Group sizes, quantification and statistical analysis are all appropriate and the methods are clearly described.

While SMC/P-specific loss of Oct4 does not affect vascular structure in healthy animals, the number of reporter-expressing cells (used as a proxy for Oct4 deletion, see point 6 below) was significantly reduced after injury in both models. This is relatively unexplored (point 1 below). The authors conclude that the defective angiogenesis results from reduced migration in Oct4-mutant SMC/P. However, convincing evidence for a migration defect is not presented (see point 2 below). Analysis of RNAseq data from in vitro cultured cells exposed to hypoxia suggests that Oct4 directly regulates Slit3/Robo2 in SMC/Ps, but this is not functionally tested (see point 3 below).

This work is an important demonstration that Oct4 reactivation in somatic cells plays a role in angiogenesis. However, insight into the cellular phenotype and/or molecular mechanism would increase the general interest in this observation.

Major points

1. The reduced number of labelled cells after injury suggests that Oct4 deletion either decreased cell survival or proliferation (or both). This apparent hypersensitivity to damage/hypoxia is an important finding, which may explain the observed defects in angiogenesis. It would be informative to know whether this is due to increased apoptosis and/or reduced proliferation at early timepoints after injury. In the corneal burn model, apoptosis is scored at day 2 and proliferation at day 5, whereas the reduced cell number is evident at all timepoints analysed (from day 3).

We thank the reviewer for this insightful analysis. We agree with the reviewer that we did not adequately address the potential roles of apoptosis and/or proliferation to the reduced eYFP+ cell density at early time points after injury. We have now provided much more extensive analyses of both apoptosis and proliferation at early time points after injury.

Our previous data indicated no difference in eYFP+ cell apoptosis, as measured by TUNEL staining, at day 2 post-burn between Oct4 WT and KO corneas. We performed similar analysis at day 5 post-burn and again observed no significant difference in eYFP+ cell apoptosis between genotypes. This data is now graphed in **Supplemental Fig. 4b**. To determine whether altered eYFP+ cell death occurs prior to day 2 post-burn, we performed corneal burn surgeries on a new cohort of mice (n=10 WT, 11 KO) and harvested at day 1 post-burn. We quantified the extent of eYFP+ cell apoptosis using TUNEL and found that the amount of apoptosis was significantly increased in Oct4 KO corneas relative to WT corneas. Therefore, loss of Oct4 in eYFP+ cells results in increased eYFP+ cell death at 1 day post-corneal burn. We have added this new data as **Figure 3i** of the revised manuscript. This data demonstrates that Oct4 plays a critical pro-survival/anti-apoptotic role in SMC-P, such that Oct4 loss in eYFP+ leads to increased eYFP+ cell death at 24 hours post-corneal burn. This at least partially explains the decreased eYFP+ cell density in Oct4 KO corneas observed following corneal burn (**Figure 3g; Supplemental Figure 4a**).

To determine whether altered proliferation of eYFP+ cells might also contribute to the decreased eYFP+ cell density observed in Oct4 KO corneas, we assessed proliferation of eYFP+ cells at day 2 post-corneal burn by staining for Ki67. However, we found no significant difference in Ki67+ staining across genotypes. This new data, along with our previous data showing no changes in eYFP+ cell proliferation at day 5 post-burn, is presented as **Supplemental Figure 4c** of the revised manuscript.

Taken together, the above results provide evidence that the reduction in eYFP+ cell density observed in Oct4 KO mice by day 3 post-burn is due, at least in part, to increased eYFP+ cell death, with no detectable differences in eYFP+ cell proliferation. We have incorporated all of this new data into our interpretation of the overall phenotype, reflected in the text (p. 11).

We also assessed changes in EC apoptosis and proliferation at early time points after injury that could result from loss of Oct4 in eYFP+ cells. We measured EC apoptosis using TUNEL at day 5 post-burn but found no significant difference in TUNEL+ staining across genotypes. This has been combined with our previous data showing no difference in TUNEL staining at day 2 post-burn and is now graphed in **Figure 5f**. We also assessed proliferation of CD31+ cells at day 2 post-corneal burn by staining for Ki67. However, we found no significant difference in Ki67+ staining across genotypes. This data, along with our previous data showing no change in EC proliferation at day 5 post-burn, is presented in **Figure 5g** of the revised manuscript.

2. In the corneal burn model, the number of eYFP+ cells is reduced at all timepoints and all distances in Oct4-mutants relative to controls. It is therefore not surprising that the number of labelled cells that have migrated is also reduced. The conclusion that migration is affected should either be toned down or substantiated for example with cell biology evidence (for example changes in cytoskeletal arrangement) or elaboration of the Slit3 phenotype (see point 3).

In retrospect, we agree with the reviewer that our original submission over-stated the role of migration in our observed phenotype without appropriately investigating and discussing alternative cellular processes, including apoptosis and proliferation. Indeed, as shown above, Oct4 knockout also leads to increased cell death of eYFP+ cells at day 1 post-burn. In the revised manuscript, we are more conservative with our interpretation of migration and have also added text (p. 11) indicating that altered cell death contributes to the observed phenotype.

In addition to the increased eYFP+ cell death in Oct4 KO corneas at 1 day post-burn, it is possible that other cell processes are affected following SMC-P loss of Oct4 which could also contribute to the decreased eYFP+ cell density observed in Oct4 KO corneas. Another possible mechanism leading to decreased eYFP+ cell density is reduced cell-cell and/or cell-matrix adhesion. In theory, this could be observed as increased shedding of eYFP+ cells in to lacrimal gland secretions. Although not directly suggested by the reviewer, it was inspired by the reviewer's suggestion to investigate alternative mechanisms contributing to our phenotype. To test this hypothesis, we conducted a lavage of the mouse orbit to collect lacrimal secretions at multiple time points (1.5 hr, 6 hr, 24 hr) post-burn, around the window where we observed increased TUNEL staining in Oct4 KO corneas. Lavage was conducted with 50 μ l of sterile saline, carefully pipetted around the orbit of the eye. We then plated lacrimal gland secretions and imaged for native eYFP expression using an inverted fluorescence microscope. We did not detect any eYFP+ cells in lacrimal fluid of either SMC-P Oct4 KO or WT mice, suggesting that loss of eYFP+ cells in tears is not a significant contributor to our observed phenotype. However, we cannot rule out that this may have occurred but not been detectable given the inherent limitations of the sampling method.

3. The functional relevance of the proposed Oct4 target Slit3 is not at all evident. Firstly, the verification of reduced levels of Slit3 in vivo is not convincing. Since

there is a large reduction in the number of recombined (eYFP+) cells in Oct4-mutant animals and Slit3 is expressed in SMC/P's it is not surprising that Slit3 "pixel intensity" is reduced (Figure 7e). Quantification should be limited to eYFP+ cells (if this is already the case, it should be clearly stated in the legend).

We thank the reviewer for providing this astute observation. The original quantification was, indeed, not limited to eYFP+ cells. To address this concern, we re-analyzed this data, normalizing the Slit3 pixel density to eYFP pixel density, to control for the decreased eYFP+ cell population in Oct4 KO tissues. We now report in **Figure 7e** that a decrease in Slit3 is observed exclusively in eYFP+ cells.

Secondly, no evidence that Slit3 is a target of Oct4 or that its expression is sensitive to hypoxia in SMC/Ps is provided.

We apologize that our original submission was not clear in explaining our data and hope that we can clarify now. As reported in Figure 7a, we analyzed our existing RNA-Seq data on SMC cultured under normoxic versus hypoxic (1% O₂) conditions to determine genes that were differentially expressed between these two conditions in both WT and KO SMC. Slit3 was identified as part of this analysis, meaning Slit3 transcript levels are regulated by hypoxia. We then identified which of these hypoxia-regulated genes were differentially expressed in Oct4 WT versus KO SMC. We found that Slit3 transcript levels were reduced in Oct4 KO SMC relative to Oct4 WT SMC. We then used qRT-PCR to validate that Slit3 transcript levels were reduced in Oct4 KO SMC relative to Oct4 WT SMC. This data indicates that Slit3 expression is sensitive to hypoxia and is dependent on Oct4 but as indicated by the reviewer does not show that it is a direct target of Oct4. We have added text to clarify this both in **Figure 7a** itself and in the results section describing Figure 7a.

Finally, it is not clear that reduced levels of Slit3 is even related to the observed angiogenesis defect. This could be examined in vitro or by virus-mediated overexpression in vivo and could potentially strengthen the claims that this is a potential mechanism of impaired migration.

We again thank the reviewer for suggesting another means to improve the manuscript. To attempt to model migration and/or proliferation of SMC and EC into the cornea following corneal burn, we performed *in vitro* scratch wound assays in cultured monolayers of SMC and EC, with or without Slit3 recombinant protein. For both cell types, we found that Slit3 increased the number of cells in the scratch wound, indicating that Slit3 promotes migration and/or proliferation in both SMC and EC. We have added this data as **Figure 7f-g**. Taken together, results suggest that reduced levels of Slit3 following loss of Oct4 at least partially contributes to impaired angiogenesis.

Further, while we agree that gain-of-function experiments to rescue the Oct4 knockout phenotype could be valuable, and seriously considered them in the past, there are a number of severe limitations that we believe make the proposed experiments highly unlikely to succeed: 1) Slit-Robo guidance signaling depends on secreted Slit

ligands establishing and maintaining temporally and spatially regulated gradients that are extremely dynamic during vascular remodeling, as cells are constantly changing their position in space and in relation to one another. While these gradients are autocrine and paracrine, acting in close coordination to the cell, precise details of Slit-Robo signaling have not yet been established in the literature. Restoration via viral-mediated overexpression would fail to re-establish the normal, highly regulated Slit3 gradients established by SMC-P and other cell types throughout angiogenesis. Rather, this might further disrupt Slit3 gradients and in turn, contribute to aberrant angiogenesis. In addition to direct disruption of Slit3 gradients, other Slit gradients including Slit1 and Slit2 may also be disrupted as cells attempt to compensate for altered Slit3 levels. 2) Multiple other cell types in the cornea and hindlimb, including EC (Zhang et al, Blood, 2009) and monocytes (Geutskens et al, Journal of Immunology, 2010), migrate in response to Slit3. We predict that overexpression of Slit3 would also result in substantially altered endothelial and monocyte migration, which would confound our results. 3) While we believe decreased Slit3 expression is related to our angiogenesis phenotype, we appreciate that our phenotype is the cumulation of numerous genes altered in response to a loss of Oct4, as evidenced by our RNASeq analyses which identified thousands of genes up or down-regulated following loss of Oct4. Consequently, it is extremely unlikely that overexpression of Slit3 alone would be sufficient to restore our *in vivo* phenotype.

Alternatively to analysis of Slit3 functional relevance, more insight into the cellular phenotype and Oct4 targets could be gained from transcriptomic profiling of eYFP+ cells isolated from animals after injury.

This is an insightful suggestion and one which we attempted to address through single-cell RNASeq analysis. Due to the increased cell death observed in eYFP+ cells of Oct4 KO corneas at day 1 post-corneal burn, we predicted this would be an ideal time point at which to analyze the eYFP+ cell transcriptome. To this end, we separately pooled 8 Oct4 WT corneas and 9 Oct4 KO corneas at 24 hours post-burn. We used SYTOX blue, a cell impermeant dye, and native eYFP expression to isolate eYFP+ cells with intact cell membranes. Despite pooling corneas, the eYFP+ cell yields were less than 2000 cells and the concentration of cDNA following RT-PCR was too low to confidently proceed with 10X NexGen sequencing. To demonstrate our attempt, we have included the flow plots below:

WT 24 hours post-burn:

KO 24 hours post-burn:

For comparison, we have also included eYFP+ cells collected from unburned corneas, which demonstrates adequate eYFP+ cell yield:

4. As discussed, the reduced number of labelled cells in Oct4-mutant animals could result in compensatory activation of progenitor cells in the vascular wall. It would be important to assess the extent of non-labelled cells acquiring a SMC/P phenotype, for example by staining for pericyte markers (Ng2).

This is a very interesting suggestion. To address the reviewer's concern, we stained whole eye cross-sections collected at day 5 post-burn for DAPI, eYFP, NG2, and CD31 and investigated whether eYFP-/NG2+ cells populated CD31+ EC tubes devoid of eYFP+ cell investment. We have shown representative images of our

observations in our new **Supplemental Fig. 5**. Oct4^{SMC-P WT/WT} corneas contained CD31+ cells invested with eYFP+ cells both in proximal and distal regions of the remodeling corneal vasculature. Most of these eYFP+ cells were also NG2+ (**Supplemental Fig. 5a**). In Oct4^{SMC-P Δ/Δ} corneas, CD31+ EC tubes in distal regions of the remodeling corneal vasculature lacked eYFP+ cell investment, consistent with observations in Figure 3. Strikingly, however, these EC tubes were invested, at least partially, by eYFP-/NG2+ cells. (**Supplemental Fig. 5b**) Confocal imaging of 3 WT and 3 KO mice revealed that these phenomena occur consistently.

Although these observations suggest that eYFP-/NG2+ pericytes may at least partially compensate for the loss of eYFP+ cells following Oct4 knockout, we present this data with a couple of important caveats: 1) While NG2+ cells enwrapping CD31+ cells is highly suggestive these cells are SMC-P, NG2 expression is not unique to SMC-P. In fact, **Supplemental Fig. 5b (box 4)** shows NG2+ cells, likely nerve cells, that are not wrapped around CD31+ EC. Furthermore, NG2 expression is dynamic, particularly in a remodeling cornea. If cells up or downregulate NG2 expression during the course of angiogenesis, immunostaining is entirely dependent on the time point of analysis. To more adequately quantify the extent of compensation by NG2+ SMC-P following Oct4 knockout, we would need to utilize a second, Cre-independent lineage tracing system specific for NG2. Unfortunately, such a mouse does not currently exist and would take several years to generate and even if generated would likely yield equivocal data due to the promiscuity of NG2 expression. 2) We cannot rule out the possibility that there are Myh11-expressing cells that fail to undergo recombination at the ROSA locus during tamoxifen administration. Consequently, these cells would not be labeled with eYFP. This would result in false-negative eYFP-/NG2+ cells of Myh11+ origin investing CD31+ tubes. If this occurred, it would be a relatively rare population, but if these same cells also failed to recombine at the Oct4 locus within Oct4 KO tissue, they would presumably have a survival advantage and could selectively expand within tissue, further confounding results. With these limitations in mind, compensation by eYFP- cells clearly does occur in Oct4 knockout corneas but fully addressing the extent to which this occurs and the origin of such cells would require multiple lineage tracing systems coupled with our Myh11-CreERT2 system and Oct4 knockout. We have added text to the discussion, highlighting similar caveats regarding this data (pp. 14-15).

Minor points

5. It is not clear whether the baseline quantification of eYFP+ cells shown in Figure 2e was performed immediately before injury. If not, quantification immediately before injury should be included to eliminate the possibility that Oct4 deletion affects cell survival or turnover.

We apologize for the lack of clarity. The quantification in Figure 2e was performed immediately before injury. We have clarified this in the figure legend.

6. Efficient Oct4 deletion in reporter-expressing cells is not verified. Recombination efficiency is cell type and locus dependent (Vooijs and Berns,

10.1093/embo-reports/kve064) and it is important to check that reporter expression reflects deletion of both Oct4 alleles at least at the genomic level.

We would like to thank the reviewer for this great suggestion. To address this point, we collected a number of tissues from tamoxifen-treated Oct4^{SMC-P WT/WT} and Oct4^{SMC-P Δ/Δ} mice, including aorta, liver, lung, diaphragm, skeletal muscle, and blood. We isolated DNA and ran PCR using primers specific for the 5' flox band (primer set A) of the non-recombined Oct4 locus or primers specific for the recombined Oct4 locus (primer set C) (**Supplemental Fig. 1a**). We detected a recombined band in all tissues from Oct4^{SMC-P Δ/Δ} mice, except blood, which we previously showed does not contain eYFP+ cells (Shankman et al, Nature Medicine, 2015). We did not detect any Oct4 recombination in tissues from Oct4^{SMC-P WT/WT} mice (**Supplemental Fig. 1b**). Additionally, we sorted eYFP+ and eYFP- cells from calf muscle of Oct4^{SMC-P WT/WT} and Oct4^{SMC-P Δ/Δ} mice and ran PCR using primer set A or primer set C. We observed Oct4 recombination only in eYFP+ cells isolated from Oct4^{SMC-P Δ/Δ} mice, not in eYFP+ cells isolated from Oct4^{SMC-P WT/WT} mice or in eYFP- cells. Taken together, this demonstrates that Oct4 recombination occurs in eYFP+ cells following tamoxifen injection in Oct4^{SMC-P Δ/Δ} mice.

7. It appears that dextran leakage has already occurred at the start of the supplementary videos. Please specify when the recordings were made relative to the injection.

The reviewer is correct in observing that dextran leakage has occurred at the start of the supplementary videos. To conduct intravital imaging of perfusion, we inject dextran retro-orbitally in the contralateral eye and then immediately position the mouse for intravital imaging. Although this process takes less than three minutes, leakage from the limbal vessels has already started. This has been clarified in our methods.

8. Individual data points should be indicated in bar graphs (journal policy).

We have updated bar graphs to include individual data points.

REVIEWERS' COMMENTS:

Reviewer #3 (Remarks to the Author):

The revised manuscript addresses all key concerns raised. I have no further comments.